# Dynamics of activation in the voltage-sensing domain of *Ciona intestinalis* phosphatase Ci-VSP

Spencer C. Guo [1,2], Rong Shen[3], Benoît Roux [1,3,4,5] & Aaron R. Dinner [1,2,4,5]

The *Ciona intestinalis* voltage-sensing phosphatase (Ci-VSP) is a membrane protein containing a voltage-sensing domain (VSD) that is homologous to VSDs from voltage-gated ion channels responsible for cellular excitability. Previously published crystal structures of Ci-VSD in putative resting and active conformations suggested a helical-screw voltage sensing mechanism in which the S4 helix translocates and rotates to enable exchange of salt-bridge partners, but the microscopic details of the transition between the resting and active conformations remained unknown. Here, by combining extensive molecular dynamics simulations with a recently developed computational framework based on dynamical operators, we elucidate the microscopic mechanism of the resting-active transition at physiological membrane potential. Sparse regression reveals a small set of coordinates that distinguish intermediates that are hidden from electrophysiological measurements. The intermediates arise from a noncanonical helical-screw mechanism in which translocation, rotation, and side-chain movement of the S4 helix are only loosely coupled. These results provide insights into existing experimental and computational findings on voltage sensing and suggest ways of further probing its mechanism.

Voltage sensing in membrane proteins is an essential biophysical phenomenon that underlies many physiological processes. Experiments and models suggest that voltage-sensing proteins respond to changes in the transmembrane potential through the movement of a transmembrane helix with several positively charged amino acids (primarily arginines)[1-3], but a detailed understanding of the mechanism at the atomic level remains lacking. Despite the availability of several structures of voltage-gated ion channels (VGICs) in their active states[4,5], there are comparatively few structures of VGICs in their resting states without the perturbing effects of engineered crosslinks (metal binding sites and disulfide bridges) or toxins[6-11].

Like VGICs, the voltage-sensing phosphatase from *Ciona intestinalis* (Ci-VSP) exhibits sensing currents in response to changes in transmembrane potential, as well as voltage-regulated enzymatic activity[12-16]. Furthermore, crystal structures of the voltage-sensing domain (VSD) from Ci-VSP in conformations thought to represent the resting (down) and active (up) states are available[17]. The down state is occupied by the wild-type protein at 0 mV applied voltage, whereas the up state is obtained by mutating a basic residue to an acidic one (R217E), which shifts the voltage dependence. These structures present a unique opportunity for elucidating a mechanism of voltage-sensing in atomic detail using molecular dynamics (MD) simulations.

[1]Department of Chemistry, The University of Chicago, Chicago, IL 60637, USA. [2]James Franck Institute, The University of Chicago, Chicago, IL 60637, USA. [3]Department of Biochemistry and Molecular Biology, The University of Chicago, Chicago, IL 60637, USA. [4]Institute for Biophysical Dynamics, The University of Chicago, Chicago, IL 60637, USA. [5]These authors jointly supervised this work: Benoît Roux, Aaron R. Dinner. ✉e-mail: roux@uchicago.edu; dinner@uchicago.edu

Consistent with structures of a number of VGICs[4,5,16], Ci-VSD consists of four transmembrane helices, labeled S1–S4 (Fig. 1). The voltage-sensing response is mediated by five arginines (R217, R223, R226, R229, and R232) spaced by three residues along S4[3,12,16–18]; these are the analogs of the gating arginines in ion channels. These arginines form salt bridges with acidic side chains acting as countercharges, which are located on helices S1–S3: D129, E183, and D186. Studies that manipulate these interactions and corresponding ones in other VSDs indicate that these interactions are essential for the voltage-sensing response[6,7,19–21]. Finally, there is a group of three highly conserved, bulky hydrophobic residues, I126, F161, and I190, frequently referred to as the "hydrophobic plug" or "gasket", which serve as a charge transfer center for the sensing arginines by focusing the transmembrane electric field into a small region[22–24]. The structures of the down and up states and simulations suggest that the S4 helix undergoes a helical-screw mechanism in discrete steps ("clicks") that couple a translocation of about 5 Å to a rotation of 60° around the helix axis[17,21]. This enables exchange of salt bridge partners between neighboring arginines. At the same time, the S1–S3 helices differ little in between the down and up states. There is evidence for states that are one click in each direction beyond the down and up states (down− and up+, respectively)[13,21,25].

Although there have been previous MD studies of VSDs[21,26–33], direct simulation of the transition between resting and active states at physiological voltages remains computationally intractable. In ref. 32, the authors were able to recapitulate the gating cycle of the Kv1.2/2.1 chimera by applying hyperpolarizing or depolarizing potentials of magnitudes over 300 mV. In these simulations, the S4 helices translocated 15 Å and rotated about 120° while the sensing arginines moved past a central hydrophobic residue in discrete steps. However, such strongly applied voltages can impact the mechanism, and the statistical significance of these observations remains unclear from the few transitions observed during milliseconds of simulation time. Enhanced sampling methods have been used to compute free energy landscapes for Ci-VSD[21] and the VSD of the Kv1.2 channel[33], but such calculations do not directly provide information about kinetics. Furthermore, they assume a priori that certain coordinates are important for describing the mechanism.

Here, we dissect the down-up transition of Ci-VSD at physiological voltages by analyzing 415 $\mu$s of unbiased MD simulations. By applying recently introduced computational methods[34,35] to obtain statistics of transitions between the down and up states of Ci-VSD[36], we elucidate the kinetics and microscopic events underlying activation. Despite capturing the apparent two-state behavior of the displacement charge, the analysis reveals multiple intermediates and deviations from a canonical helical-screw mechanism. These findings provide insights into the origins of the complex kinetics of activation measured for wild-type and mutant Ci-VSPs, which we set in the context of results for other voltage-sensing proteins. We conclude by suggesting experiments that can test the computational predictions.

## Results

Our goal is to compute kinetic statistics for the transition between the down and up states and then relate these statistics to specific microscopic events. We rely on a framework called the dynamical Galerkin approximation (DGA) that assumes only that the dynamics are stochastic and Markovian after a relatively short time[34,35]. The essential idea is that long-time statistics (described below) of the down-up transition can be estimated by combining information from a dataset of short unbiased MD trajectories, each of which samples a portion of the VSD activation event. In other words, we can learn the mechanism of the transition even if no single trajectory connects the down and up states, as long as conformations belonging to the transition region between them are adequately represented in the dataset (discussed further in Methods). This approach dramatically reduces the computational cost because we do not need to wait for fluctuations that give rise to a trajectory transitioning between the down and up states, let alone the many that would be needed to achieve statistical confidence. We exploit the specialized Anton 2 supercomputer[37] to generate a dataset of 415 $\mu$s of aggregate simulation at zero (depolarized) membrane potential. The dataset consists of short trajectories initiated from starting points distributed between the down− and up+ states (homology models that shift the sensing arginines one click past the down and up states[17,21]).

We describe three statistics used in this study and how we use them to learn the molecular mechanism. We define them mathematically and discuss how we compute them in Methods.

- *Equilibrium probabilities of states* define relative free energies, which can be used to compute equilibrium averages and potentials of mean force (PMFs) along selected coordinates.
- The *committor* is the probability of completing the transition (i.e., proceeding to the up state before returning to the down state).
- The *reactive current* describes the fluxes between states.

These statistics provide key insights into complex conformational transitions. For example, the PMF allows us to visualize metastable (i.e., long-lived) states and the barriers separating them. The committor, because it varies monotonically between zero at the reactant state and one at the product state, represents a natural coordinate for tracking the progress of a conformational transition and can be used to identify the molecular features that correlate with that progress. Finally, the reactive current yields information about how molecular features change during a conformational transition. We use it to visualize transition pathways and quantify their relative contributions.

We first connect our simulations to experimental measurements via the displacement charge. Next, we show how the committor provides additional microscopic insight into the kinetics, using a sparse regression procedure to identify a small set of molecular collective variables (CVs) useful for modeling activation. Projections of the reactive currents and free energies onto these CVs enable us to characterize the transition pathways and thermodynamic forces stabilizing them. The overall workflow is summarized in Fig. 2.

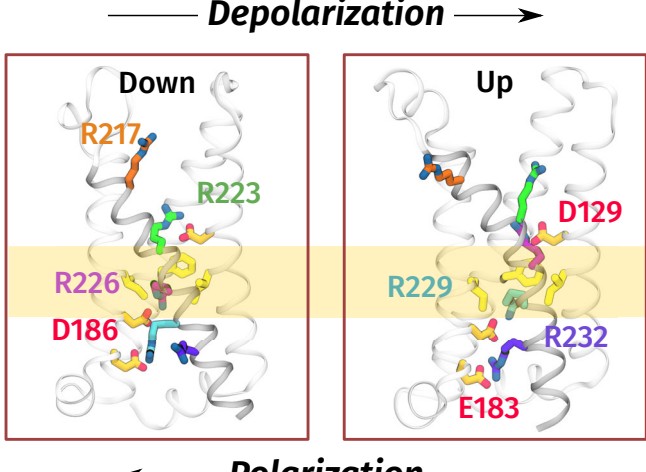

**Fig. 1 | Structure of Ci-VSD.** Left, down (PDB: 4G80) and right, up (PDB: 4G7V) states. The side chains of sensing arginines on the S4 helix (gray) which participate in voltage-sensing, R217 (orange), R223 (lime), R226 (magenta), R229 (cyan), and R232 (purple), as well as residues that form salt bridges with them (D129, E183, D186) are rendered as sticks. Nitrogen and oxygen atoms on these side chains are colored blue and red, respectively. Hydrophobic plug residues (I126, F161, and I190) are highlighted in yellow.

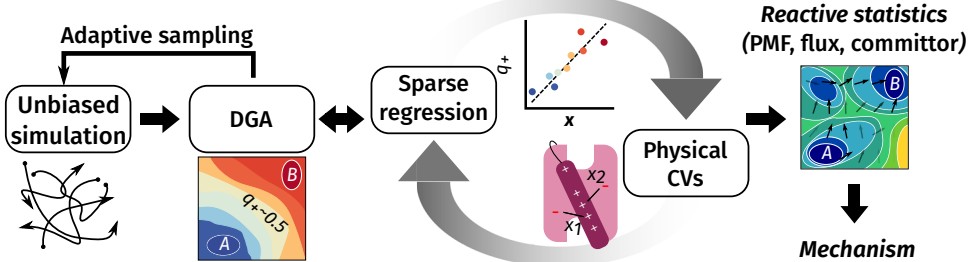

**Fig. 2 |** Computational workflow for this study.

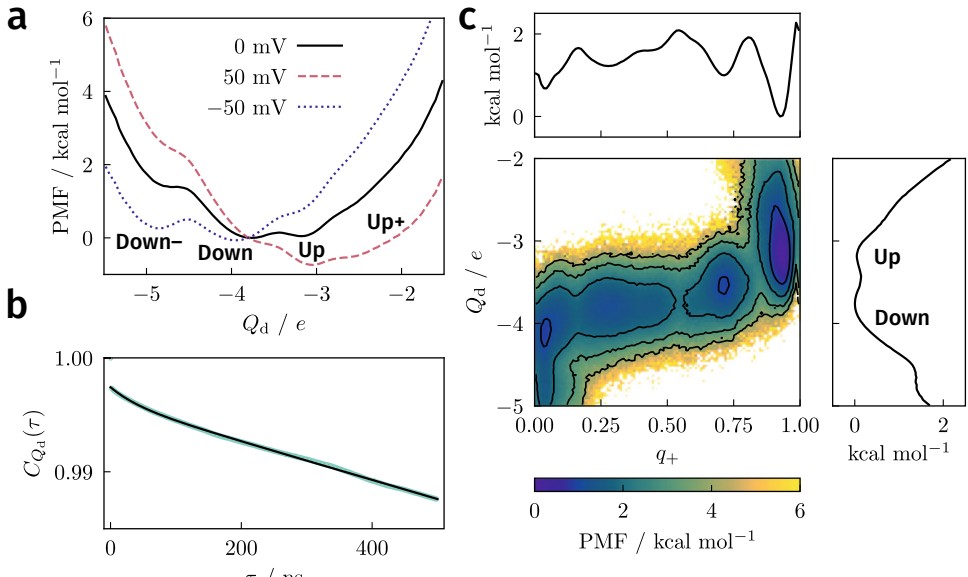

**Fig. 3 | Displacement charge. a** PMF computed from DGA along $Q_d$. The free-energy profiles at nonzero voltages were obtained by adding $Q_d V$ to the profile obtained under 0 mV (note that differences in free energies between states depend only on their relative rather than absolute $Q_d$ values). The down and up states correspond to $Q_d$ values of $-4.2 \pm 0.2\ e$ and $-3.3 \pm 0.2\ e$, respectively. **b** Time correlation function $C_{Q_d}(\tau) = \langle Q_d(0)Q_d(\tau)\rangle / \langle Q_d(0)Q_d(0)\rangle$ of the displacement charge.

Green circles were computed from short trajectories by reweighting to equilibrium. Solid line represents a biexponential fit with time constants of 59 $\mu$s and 0.04 $\mu$s. **c** PMF projected onto $q_+$ and $Q_d$ and along each individual coordinate (above and right). Contours on the two-dimensional PMF are drawn every 1 kcal/mol. Source data are provided as a Source Data file.

## The committor reveals intermediates hidden within a single "click" of the displacement charge

The displacement charge $Q_d$ represents the energetic coupling of the membrane protein system with an external potential[38]. The change in displacement charge upon a conformational transition of the voltage-sensor, $\Delta Q_d$, also known as the sensing charge (or, in VGICs, the gating charge), can be measured with voltage-clamp electrophysiology[3,39]. Ci-VSD was previously found[12,17,21] to transfer approximately 1.1 $e$ in going from the down state to the up state. Our calculations yield a sensing charge of $\Delta Q_d = 0.9 \pm 0.3\ e$, in agreement with previous estimates.

Using DGA, we computed the average $Q_d$ as a function of the S4 helix translocation and rotation, and it exhibits discrete values (Supplementary Fig. 1a), consistent with the idea that the S4 helix moves by discrete "clicks" (these results can be compared directly with Fig. 1B of ref. 21). In further support of this idea, the PMF as a function of $Q_d$ (Fig. 3a, black line) exhibits minima at $Q_d$ values corresponding to the down and up states; these minima are nearly identical in free energy and are separated by a barrier of less than 0.5 kcal/mol, suggesting both states may be populated in the absence of an applied potential. Although the simulations are performed at 0 mV, we can mimic the effect of an applied potential, $V$, by adding a linear term $Q_d V$ to the free energy profile[38]. Relatively mild hyperpolarizing or depolarizing voltages on the order of $\mp$50 mV are

sufficient to tilt the PMF in favor of the down (and down−) or up (and up+) states, respectively (Fig. 3a).

To connect with measured kinetics[13,14,40–43], we computed an equilibrium time-correlation function for $Q_d$ (Fig. 3b) and fitted it to a biexponential function with slow and fast time constants of approximately 59 $\mu$s and 0.04 $\mu$s, respectively. The slow time constant is consistent with our ~100 $\mu$s estimate of the mean first passage time between the down and up states (Supplementary Methods and Supplementary Fig. 2). These values are about an order of magnitude faster than the fast component of fluorescence relaxation of dye-labeled VSDs, which tracks the charge movement (sensing current)[13,40–43]. However, the experimental studies were mainly performed with full-length Ci-VSP containing the phosphatase domain and linker, which slows down the voltage-activation process[40]. With this in mind, our results suggest both the equilibrium averages and the kinetics are consistent with available data and previous simulations for Ci-VSD, so we now turn to dissecting the microscopic dynamics during activation.

As a first step toward this end, we computed the committor ($q_+$) for all structures in our dataset. To recap, the committor is the probability that trajectories initialized from a conformation lead to the up (i.e., active) state before the down (i.e., resting) state, so the down state corresponds to $q_+ = 0$ and the up state corresponds to $q_+ = 1$. We used

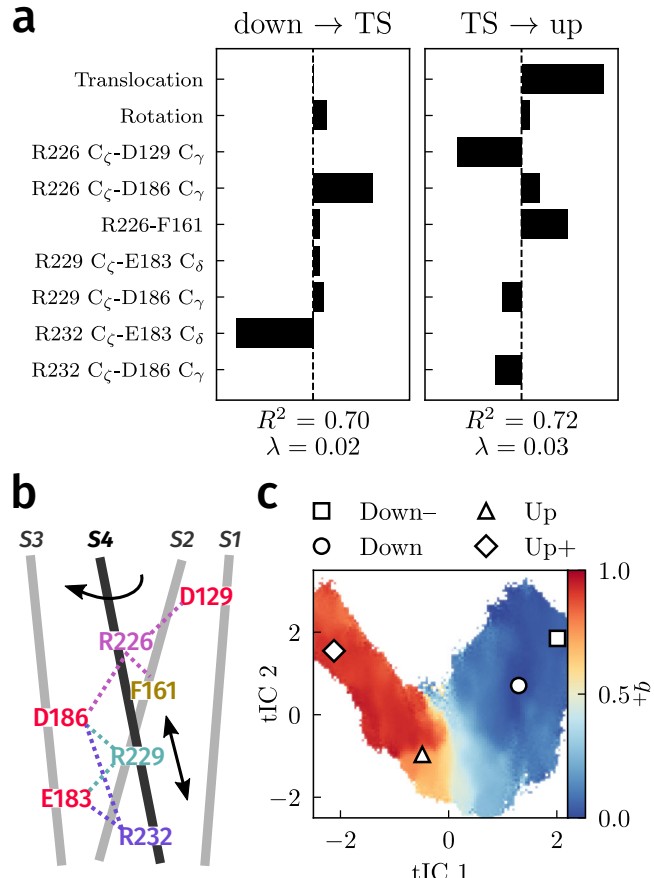

**Fig. 4 | Dependence of the committor on collective variables (CVs).**
**a** Coefficients computed from LASSO regression for points with $q_+ \leq 0.5$ and $q_+ \geq 0.5$.
**b** Schematic of CVs (S4 helix variables and intramolecular distances) used for regression. **c** Committor projected onto the first two nontrivial IVAC coordinates (tICs) constructed from the CVs in (**a**). Source data are provided as a Source Data file.

the committor to compute a two-dimensional PMF (Fig. 3c), revealing a qualitatively different picture than the PMF of the displacement charge alone. There are minima at $q_+ \approx 0.25$, 0.50, and 0.75, all with $Q_d$ between $-3.5\,e$ and $-4\,e$. The simulations thus suggest that, while the displacement charge dynamics are essentially two-state, the microscopic dynamics are five-state. Moreover, the highest barrier is about 1.5 kcal/mol relative to the down state. To understand the differences between the PMFs and to characterize the intermediates and barriers between the down and up states, we sought to identify molecular features that correlated with the committor.

### Sparse modeling of the committor identifies physically interpretable variables that describe the transition

Following previous work[44,45], we sought to identify combinations of physically intuitive CVs that can describe the dynamics by modeling the committor in terms of a small number of interpretable molecular features (Sparse regression). Out of the 60 inter-residue distances used as input to the committor (Basis set construction), we manually selected several to track the movements of the side chains of the sensing arginines (R217, R223, R226, R229, and R232), as well as those of their acidic salt-bridge partners. These were distances between the sensing arginine $C_\zeta$ atoms and the $C_\gamma/C_\delta$ atoms of D129, E183, and D186 (Fig. 1). We also used the distance between the $C_\zeta$ atom of R226 and the center of mass of F161's side chain to track the movement of R226 through the hydrophobic plug, which is thought to present a barrier between the resting and active states[23,24,31]. We chose F161 instead of

I126 or I190 because it is highly conserved among voltage-sensitive proteins (corresponding to F290 in the *Shaker* potassium channel for example)[23]. Finally, we included the S4 helix translocation and rotation, given their known role in VSD activation[21,29,32,46–48].

We initially fit the model to all points with $q_+ \in [0.2, 0.8]$. However, in analyzing the model, we found that some CVs that we expected to contribute had small or zero-valued coefficients because the committor varies non-monotonically with them (e.g., the R226-F161 distance decreases and then increases as R226 goes through the hydrophobic plug). Consequently, we divided the reaction into two stages, one from the down (reactant) state to $q_+ = 0.5$ (i.e., the transition state ensemble in the sense that conformations with $q_+ = 0.5$ have an equal likelihood of leading to the down and up states), and another from $q_+ = 0.5$ to the up (product) state (Sparse regression).

We found that nine CVs contributed significantly to at least one of the two models. They were the S4 helix translocation and rotation and distances involving R226, R229, and R232 (Fig. 4a, b). To verify that these CVs are indeed sufficient to express the committor (and hence represent a useful subspace of variables to analyze), we constructed coordinates from linear combinations of input variables using the integrated variational approach to conformational dynamics (IVAC)[49], a method that robustly identifies slow modes of a system. The committor monotonically follows the first nontrivial IVAC coordinate (Fig. 4c), suggesting that the down-up transition is fully captured using our chosen subset of variables. In addition, the committor projection on the IVAC coordinates has much lower variance compared with the projection on the S4 coordinates (Supplementary Fig. 3), reflecting the information gained from tracking the sensing arginines. Given this, we now interpret the PMF in Fig. 3c in terms of these variables. We first discuss the S4 helix translocation and rotation and then the side chain dynamics.

### S4-helix translocation and rotation are loosely coupled

The up state crystal structure corresponds to the origin of the translocation-rotation plane, and the down state crystal structure lies at approximately $(-4.5\,\text{Å}, -60°)$ (Fig. 5). We can compute the PMF along these coordinates from our dataset and compare it directly with the PMF in ref. 21 obtained using replica exchange umbrella sampling (REUS). The two PMFs are similar overall but differ slightly near the down state (Fig. 5a and Supplementary Fig. 4), and there is a prominent minimum with a depth of about 1–2 kcal/mol near $(-5\,\text{Å}, -10°)$. This minimum is also present in the REUS simulations at positive applied potentials, when up-like states are stabilized[21]. Structures in this minimum have the R226 side chain located above the hydrophobic plug. Because the sampling in these CVs is controlled directly in REUS, we expect the REUS PMF to be more accurate—DGA slightly underestimates the stability of the down state and the barrier to the up state.

We show histograms of translocation and rotation as functions of the committor in Fig. 5c. These indicate that both variables have considerable freedom in the down state ($q_+ = 0$). Prior to the transition state ($q_+ < 0.5$), translocation remains unrestricted, but rotation becomes restricted and steadily increases towards 0° with the committor. After the transition state ($q_+ > 0.5$), translocation rapidly increases to its value in the up state ($q_+ = 1$). These results are consistent with the average committor, the PMF, and the reactive currents in this plane (Fig. 5a, b). The transition state ($q_+ = 0.5$) appears as a nearly horizontal white line just below the up state, and the reactive current is in the vertical direction.

Historically, two mechanisms have been proposed for S4 helix movement: a helical-screw mechanism in which translocation and rotation advance together[46–48,50,51] and a sliding-helix mechanism in which the $\alpha$-helix converts to a narrower $3_{10}$-helix, allowing it to translocate independently of rotation[7,40]. In contrast to the helical-screw mechanism, we find that the S4 helix is relatively free to translocate back and forth prior to the transition state (Fig. 5c), and it is the

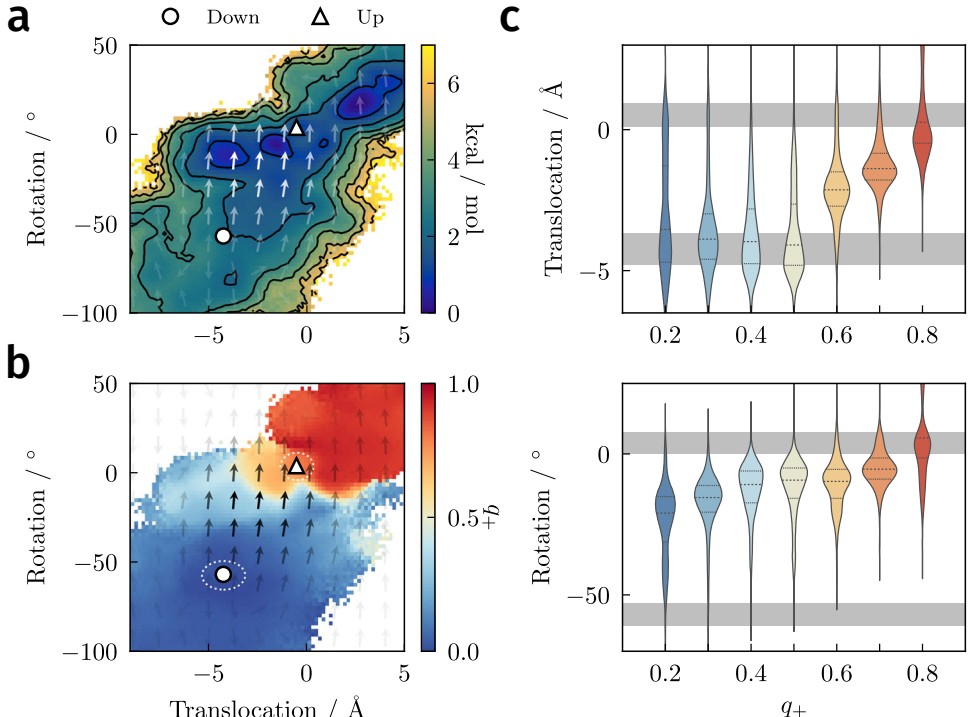

**Fig. 5 | Role of the S4 helix. a** PMF as a function of S4 helix translocation and rotation. Reactive currents are shown as a vector field with opacity indicating the magnitude of the vector. Contours are drawn every 1 kcal/mol. **b** Average committor as function of S4 helix translocation and rotation. Approximate definitions of the down and up states for the committor calculations are marked with white ellipses. **c** Distributions (violin plots) of S4 helix translocation and rotation as a function of committor. Each violin is obtained by binning structures with the indicated $q_+ \pm 0.05$ (i.e., the first violin contains structures with $q_+ \in [0.15, 0.25]$). The median and upper/lower quartiles are denoted with dashed and solid lines, respectively. Approximate values of translocation or rotation in the down and up states marked by gray bars. Source data are provided as a Source Data file.

restriction of the rotational degree of freedom that determines when the transition state is crossed. At the same time, we find little $3_{10}$-helix content to support the sliding-helix mechanism (Supplementary Fig. 5), consistent with prior analyses[21]. The mechanism that we observe combines the rotation of the helical-screw mechanism with the translocation of the sliding-helix mechanism, resembling the action of a barrel slide bolt. In the next section, we elaborate this mechanism by considering the side-chain dynamics.

**Transitions between intermediates involve exchange of arginine salt-bridge partners**
The sparse model (Fig. 4a) shows that, in addition to the S4 helix, the movements of R226, R229, and R232 relative to their salt bridge partners and the hydrophobic plug are important. We find that they can account for the structure in Fig. 3c. Moreover, we observe in our simulations that vertical translocation of the S4 helix can precede or lag behind the exchange of salt bridge partners by the sensing arginines (Supplementary Fig. 6). That is, changes in the side chains of R226, R229 (and to some extent R232), translocation, and rotation are quasi-independent. Here, we describe the successive movements from the down state to the minima at $q_+ \approx 0.25$, 0.50, and 0.75, and finally to the up state.

In the down state, R226 and R229 form salt bridges with D186 and E183, respectively (Fig. 1), while the guanidinium group of R232 forms hydrogen bonds to phosphate groups of lipids (Supplementary Figs. 7a and 8a, c), a role previously noted in studies of VSDs[26,32,52,53]. The aliphatic part of the R226 side chain is in contact with the hydrophobic plug.

The transition to the minimum at $q_+ \approx 0.25$ involves the movement of R226's guanidinium group into the plug so as to interact with the phenyl side chain of F161 (Fig. 6a, b and Supplementary Fig. 9a,b).

Although cation-$\pi$ interactions and $\pi$-$\pi$ stacking have been suggested to stabilize interactions between arginine and phenylalanine residues[23,54–56], the fixed charges in the force field do not directly represent these (quadrupole) interactions. Instead, this minimum is primarily stabilized by R226 forming a salt bridge with D129 from the hydrophobic plug (Supplementary Fig. 9c). Consistent with the discussion above, making these interactions relies on S4 helix rotation but not translocation (Supplementary Fig. 10).

The next barrier between the minima at $q_+ \approx 0.25$ and $q_+ \approx 0.50$ is very low, and R229 interacts with both E183 and D186 to varying degrees in this range of committor values (Fig. 6a and Supplementary Fig. 7a). The transition from the first intermediate minimum to the second reflects the movement of R232 away from the phosphate groups of the lipids to interact with E183 (Fig. 6a, Supplementary Figs. 7a and 10). In the intermediate near $q_+ \approx 0.5$ (which can be considered the transition state ensemble, as noted above), we find significant heterogeneity in R229's conformations: the R229-E183 distance, for example, exhibits two populations, one centered around 4 Å and another from 6 to 10 Å (Fig. 6a and Supplementary Fig. 7b). In the former, R229 is coordinated to both E183 and D186, whereas in the latter R229 is only coordinated to D186 (Fig. 6c). Accordingly, R229 forms hydrogen bonds mainly with D186, but also with E183 and lipid phosphate groups when $q_+ < 0.5$ (Supplementary Figs. 7a and 8). As discussed earlier, translocation of the S4 helix is relatively unconstrained in the intermediate stages of the transition (Fig. 5), with heterogeneity seen in transition state structures.

The transition to the minimum at $q_+ \approx 0.75$ requires R229 to dissociate from E183 and R232 to interact with D186 (Fig. 6a, d and Supplementary Fig. 10). The barrier separating the minima at $q_+ \approx 0.50$ and $q_+ \approx 0.75$ results from the exchange of the two salt bridges with R229 and R232. Furthermore, after the transition state $q_+ \approx 0.5$, S4 helix

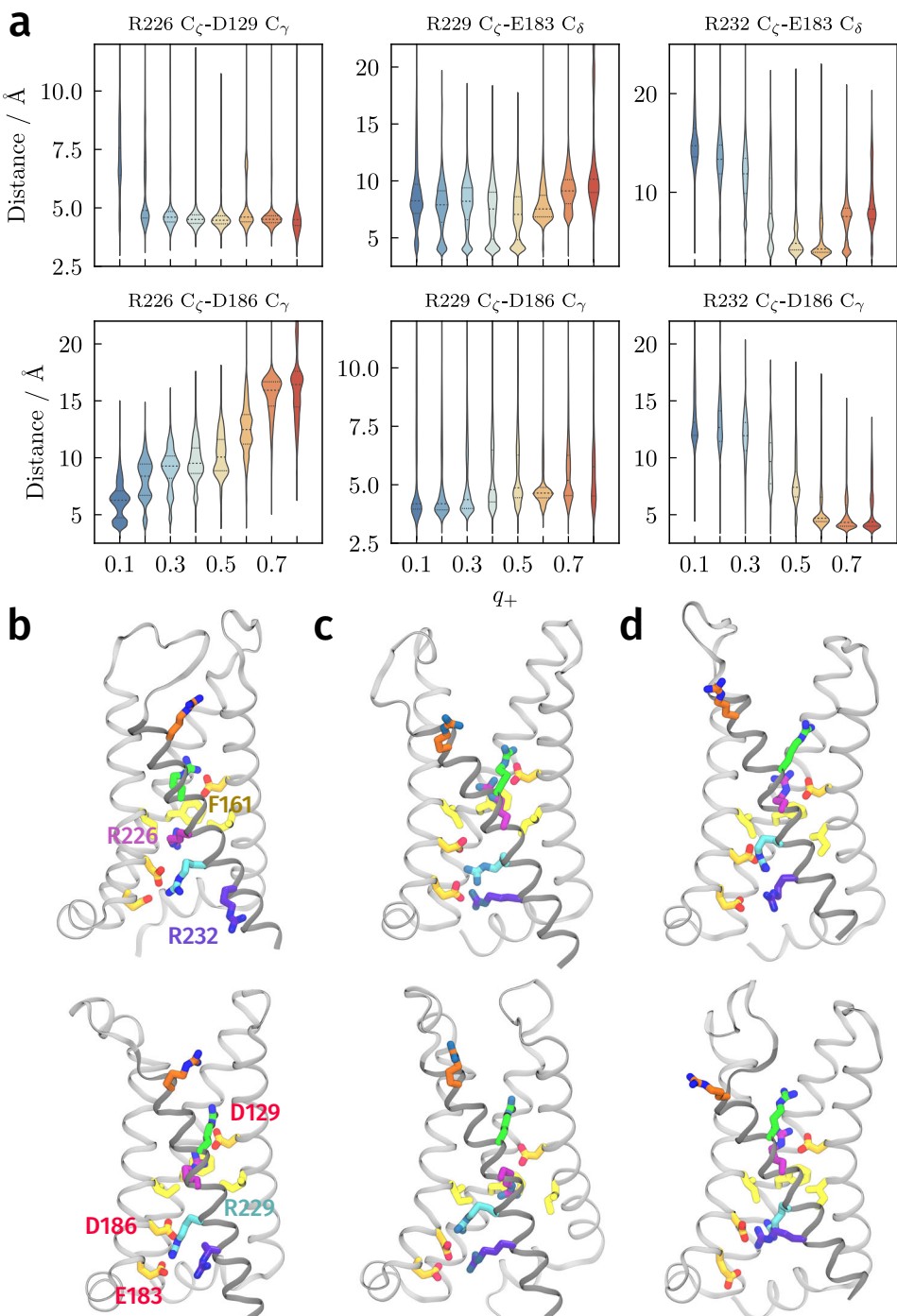

**Fig. 6 | Role of sensing arginine salt bridges and structure of the intermediates.** **a** Distribution of salt-bridge distances as a function of committor. Each violin is obtained by binning structures with the indicated $q_+ \pm 0.05$ (e.g., the first violin contains structures with $q_+ \in [0.05, 0.15]$). The median and upper/lower quartiles are denoted with dashed and solid lines, respectively. **b**–**d** Selected structures from the intermediates at $q_+ \approx 0.25$, $q_+ \approx 0.5$, and $q_+ \approx 0.75$ from $k$-medoids clustering. Source data are provided as a Source Data file.

translocation begins increasing (Fig. 5b and Supplementary Fig. 10), and the completion of this process results in the final transition to $q_+ = 1$. In some structures with high committor values, R226 extends to interact with S158 and T197 (Supplementary Fig. 7c).

## Discussion

In this paper, we applied recent advances in methods for computing kinetic statistics to elucidate the mechanism of the down-up transition of Ci-VSD. In contrast to previous studies of voltage-sensing proteins, which have focused mainly on metastable states of the voltage-sensing domain[21,29,32,33], our approach allows us to dissect the microscopic events underlying a single "click".

As summarized in Fig. 7, the down-up transition proceeds through a stepwise mechanism in which the S4 helix "walks" with its positively-charged amino acids along the negatively-charged salt bridge partners like a caterpillar on a rugged landscape (Supplementary Movie 1). The overall displacement of the helix and the rearrangements of the side chains are only loosely coordinated. The movements of R226, R229, and R232 do not follow a strict sequence, though steric interference prevents them from being totally independent. From the down state to

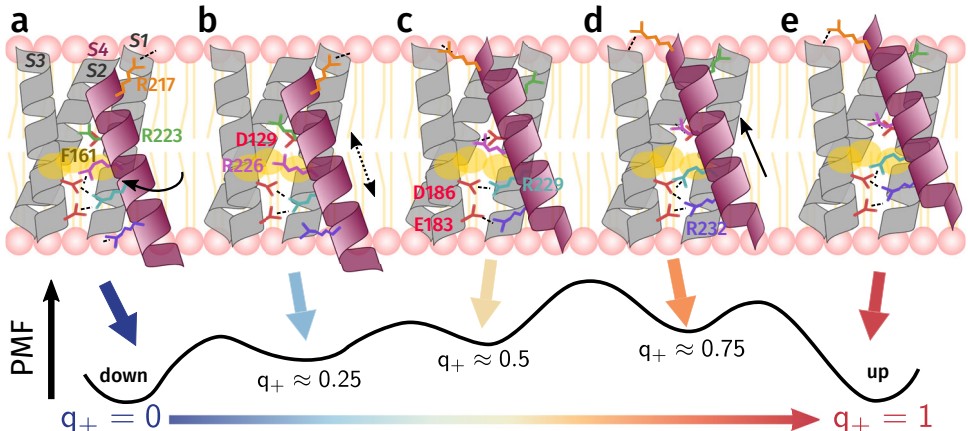

**Fig. 7 | Schematic of the mechanism of the down-up transition in Ci-VSD.**
Sensing arginines and countercharges are depicted as sticks; hydrophobic plug residues are depicted as shaded yellow circles. **a** In the down state ($q_+ = 0$), R223 forms a salt bridge with D129 above the hydrophobic plug. R226 is hydrated, while R229 interacts mainly with E183 and D186, and R232 coordinates to POPC head-groups. **b** First ($q_+ \approx 0.25$), R226 rotates upwards as the S4 helix rotates, which allows it to cross the hydrophobic plug and begin forming a salt bridge with D129. **c** Near the transition state ($q_+ \approx 0.5$), R226 interacts with D129, while R229 interacts with both E183 and D186. **d** Beyond the transition state ($q_+ \approx 0.75$), R229 detaches from E183, and R232 interacts with both E183 and D186. **e** The S4 helix translocates to its final up state position ($q_+ = 1$).

$q_+ \approx 0.25$, rotation of the S4 helix allows R226 to detach from D186, move through the hydrophobic plug, and form a salt bridge with D129 (Fig. 7a, b). At $q_+ \approx 0.5$, where the system has an equal likelihood of proceeding forward and backward, R229 delocalizes and interacts with both E183 and D186, while R232 detaches from the lipid phosphate head groups and forms a salt bridge with E183 (Fig. 7c). At $q_+ \approx 0.75$, R229 forms a salt bridge to D186 and R232 interacts with both E183 and D186 (Fig. 7d). Finally, the S4 helix must translocate to its up state position while R229 enters the plug to occupy the position of R226 in the down state (Fig. 7d, e).

Although the free-energy landscape along the displacement charge appears two-state, our committor-based analysis reveals the presence of metastable intermediates. These intermediates may help explain the multi-exponential kinetics of fluorescence relaxation measured for labeled VSDs[13,40–43]. Given that the PMFs are relatively shallow, even under applied transmembrane potentials of ±150 mV[21], there could also be contributions from transitions involving the down– and up+ states, which could introduce additional timescales. Regarding the kinetics, it is important to note that our estimates of the activation timescales are systematically too fast. This may reflect a breakdown in the Markov assumption underlying the DGA calculations, limitations of the choice of basis, insufficient sampling, and limitations of the force field. The first two could be addressed by including non-Markovian effects[57,58] in the DGA calculations or by applying nonequilibrium sampling methods[59].

Despite the potential issues, we expect that our simulations represent the microscopic structures and dynamics of Ci-VSD close to 0 mV reasonably well. We consequently expect the "caterpillar" mechanism that we describe to hold for mildly depolarizing and hyperpolarizing voltages. This mechanism represents a notable deviation from a canonical helical-screw model in that translocation, rotation, and side-chain movements are only loosely coupled. While we are not aware of a previous study that explicitly describes such a mechanism, some decoupling of the S4 translocation, rotation, and side-chain movements has been noted in passing for simulations of Ci-VSD[21] and Kv1.2[27,29].

Another striking feature of our simulations is the relatively low barrier between the down and up states in Ci-VSD compared with, for example, the barrier between the $\delta$ and $\varepsilon$ states of the Kv1.2 channel (5 kcal/mol)[33]. Given Ci-VSP's sequence divergence from VGICs and comparatively slower gating kinetics[12,17,18,24,40,60], we believe the difference in barrier height is genuine and not an artifact of different computational approaches. The lower barrier could furthermore be partially attributed to the lack of $3_{10}$ helix formation[7,21].

We expect that R226's movement through the hydrophobic plug is facilitated by its ability to make a salt bridge with D129 while in contact with the I126 and F161 side chains (Supplementary Fig. 9c). Mutagenesis experiments in Ci-VSD[61,62] and closely related voltage-gated proton channels[63] support the idea that D129 is essential for voltage-dependence. In contrast, VGICs generally have a polar but uncharged amino acid at the site corresponding to D129[16,64], preventing formation of an analogous interaction and leading to a significant dielectric barrier[65]. Perhaps relatedly, in VGICs, reducing the hydrophobicity of gating-pore residues by mutation accelerates activation upon depolarization[65], whereas in Ci-VSP kinetics are more closely correlated with the sizes of side chains of hydrophobic-plug residues[24,66]. The closer spacing of the acidic residues in Ci-VSD compared with VGICs may also explain how the arginine side chains can make salt bridges without the S4 helix needing to convert to a $3_{10}$-helix near the hydrophobic plug, as in the sliding-helix mechanism[7,21,30,67].

Experimentally, it may be possible to validate the proposed mechanism by correlating movements of a fluorescent label (e.g., attached to a G214C mutation on the S3-S4 linker) to S4 helix movement and/or displacement charge changes[13,14,25,40,62,68], or while perturbing residues implicated in key rearrangements. For example, the importance of R226 side-chain reorientation during early stages of activation suggests that limiting rotation of the $\chi_1$ dihedral angle (perhaps by introducing an unnatural arginine congener[69]) might alter the kinetics of activation or block it entirely. Other residues not analyzed in detail, such as S158 and T197[21] (Supplementary Fig. 7c), as well as W182[25,68], may modulate intermediate states of the activation process by interacting with the sensing arginines, and their role could be tested by observing the effect of mutations on activation kinetics.

Notably, the physically interpretable low-dimensional model of the committor did not require distances to the sensing arginines R217 and R223. R217's interactions with negative countercharges such as D129, D136, and D154 correlate only weakly with the committor and are heterogeneous in the transition state (Supplementary Fig. 7b). Our results are consistent with prior suggestions that R217 primarily participates in activation by modulating the electric field rather than through salt-bridge formation[18], although its role may be different in the transition between the down– and down states owing

to its proximity to D129 and the hydrophobic plug in the down-state. On the other hand, R223 tends to form salt bridges with D129. Relative to R226, R223's salt-bridge distances are less predictive of the committor and its contribution to the displacement charge is small, consistent with its exposure to solvent in the extracellular crevice of Ci-VSD. Prior studies that mutated this amino acid to cysteine or histidine[18,21] suggested that R223 may play a larger role in transitions involving the down- and up+ states of Ci-VSD. Computationally, it would therefore be interesting to study these other transitions, as well as the down-up transition in the presence of applied fields.

## Methods

We compute statistics from MD simulations using the dynamical Galerkin approximation (DGA)[34,35]. The aim of DGA is to formulate and solve an operator problem for these statistics as a linear system of equations. The theoretical framework is based on transition path theory[36] and only assumes that the dynamics are Markovian within a user defined time interval termed the lag time ($t$ below). The key advantage of this approach is that estimating the terms in the linear equations requires only short trajectories (on the order of the lag time) rather than ones recapitulating full transition paths from the down state to the up state. The main challenge is ensuring that the short trajectories in the aggregated simulation dataset adequately sample all parts of the desired transition (see Simulation details and adapative sampling procedure).

The resulting estimates of the statistics depend on many variables (the input molecular features to the basis functions) and so to interpret them, we use sparse regression to identify the most important collective variables (CVs) for describing the conformational change. Readers who wish to obtain an overview of the DGA framework without delving into the mathematical details need only skim Key statistics, Dynamical Galerkin approximation, and Basis set construction.

### Key statistics

As described in Results, we use three statistics to understand the mechanism of the down-up transition: the equilibrium probability, the committor, and the reactive current. The equilibrium distribution (or more precisely, its ratio to the sampling distribution, $w$) and the committor, $q_+$, satisfy equations of transition operators, which describe how averages of functions of the configuration evolve under the dynamics. The reactive current, $J_{AB}^{\xi}$, can be computed from the equilibrium distribution and the committor. In this section, we define the operators mathematically and present key equations for these statistics.

For a Markov process $\mathbf{X}(t)$, the transition operator $\mathcal{T}_t$ is defined as

$$\mathcal{T}_t f(\mathbf{x}) = \mathbf{E}[f(\mathbf{X}(t)) \mid \mathbf{X}(0) = \mathbf{x}],\qquad(1)$$

where $f$ is an arbitrary function of the coordinates $\mathbf{X}(t)$, and $\mathbf{E}$ denotes an expectation. The transition operator evolves the average of $f$ (over multiple random realizations of $\mathbf{X}$) for a lag time $t$, conditioned on starting at $\mathbf{x}$.

The PMF is the free energy as a function of specific coordinates. We can obtain the free energy from the negative logarithm of the equilibrium distribution $\pi(\mathbf{x})$ (which is proportional to the Boltzmann weight in the canonical ensemble). Specifically, we introduce a weight factor $w(\mathbf{x})$ that relates the sampled data distribution $\mu(\mathbf{x})$ (which need not be drawn from $\pi$) to $\pi(\mathbf{x})$:

$$\int f(\mathbf{x})\pi(\mathbf{x})\mathrm{d}\mathbf{x} = \int f(\mathbf{x})w(\mathbf{x})\mu(\mathbf{x})\mathrm{d}\mathbf{x}.\qquad(2)$$

Because $\pi(\mathbf{x})$ is the equilibrium distribution, by definition it must not change over time or, in other words, upon application of the transition operator. The weight factor can be shown[35,70] to satisfy

$$(\mathcal{T}_t)_\mu^\dagger w(\mathbf{x}) = w(\mathbf{x}),\qquad(3)$$

where $(\mathcal{T}_t)_\mu^\dagger$ is the $\mu$-weighted adjoint of $\mathcal{T}_t$. We solve this equation using DGA as described below.

The committor $q_+(\mathbf{x})$ is defined as the probability of reaching the product state $B$ (here the up state) before returning to the reactant state $A$ (here the down state), conditioned on starting at $\mathbf{x}$. In this sense, it is a perfect reaction coordinate (measure of progress) for a stochastic reaction[36]. Because of the boundary conditions $q_+(\mathbf{x}) = 0$ in $A$ and $q_+(\mathbf{x}) = 1$ in $B$, the committor is governed by a modified transition operator

$$\mathcal{S}_t f(\mathbf{x}) = \mathbf{E}\big[f(\mathbf{X}(t \wedge T_{A \cup B})) \mid \mathbf{X}(0) = \mathbf{x}\big],\qquad(4)$$

where $T_{A \cup B} = \min\{t \geq 0 \mid \mathbf{X}(t) \in A \cup B\}$ is the first time $\mathbf{X}$ enters $A \cup B$ and $t \wedge T_{A \cup B} \equiv \min\{t, T_{A \cup B}\}$. The transition operator $\mathcal{S}_t$ corresponds to terminating the dynamics when $A$ or $B$ is entered. The committor satisfies[35]

$$q_+(\mathbf{x}) = \begin{cases} \mathcal{S}_t q_+(\mathbf{x}) & \mathbf{x} \in (A \cup B)^c \\ 0 & \mathbf{x} \in A \\ 1 & \mathbf{x} \in B. \end{cases}\qquad(5)$$

Crucially, (5) holds for any lag time $t$, so in principle we can obtain $q_+$ from simulations that are much shorter than the transition time (i.e., $t \ll T_{A \cup B}$).

Finally, the reactive current measures the flux of reactive probability density, the probability of observing a reactive trajectory transitioning from $A$ to $B$[36,71]. The reactive current is most easily interpreted after projecting onto a subspace of CVs, $\xi(\mathbf{x})$, yielding a vector field $J_{AB}^{\xi}(s)$ that summarizes how reactive trajectories flow through $\xi(\mathbf{x}) = s$. It was shown previously that the projected reactive probability current satisfies[35,59]

$$J_{AB}^{\xi}(s) = \int J_{AB}(\mathbf{x}) \cdot \nabla\xi(\mathbf{x})\delta(\xi(\mathbf{x}) - s)\pi(\mathbf{x})\mathrm{d}\mathbf{x},\qquad(6)$$

where $J_{AB}$ is the reactive flux in the space of original coordinates. As in previous work, we smooth all reactive currents with a Gaussian kernel density estimate of width 1 bin.

### Dynamical Galerkin approximation

In DGA, we write the statistic of interest, such as $w$ or $q_+$, as a linear combination of basis functions $\phi_i(\mathbf{x})$ with coefficients $\mathbf{v}_i$:

$$f(\mathbf{x}) = \gamma(\mathbf{x}) + \sum_{i=1}^N \mathbf{v}_i\phi_i(\mathbf{x}).\qquad(7)$$

Here, $\gamma(\mathbf{x})$ is a "guess" function chosen to satisfy the boundary conditions, so $\phi_i(\mathbf{x}) = 0$ for $\mathbf{x} \in A \cup B$. Note that solving for the weight factor $w$, unlike $q_+$, does not require any explicit boundary conditions. After substituting the ansatz in equation (7) into the appropriate operator equation [e.g., equation (5) for $q_+$], multiplying by another basis function $\phi_j$, and integrating over the sampled distribution $\mu(\mathbf{x})$, we obtain the linear system

$$(\mathbf{C}_t - \mathbf{C}_0)\mathbf{v} = -(\mathbf{r}_t - \mathbf{r}_0),\qquad(8)$$

where

$$\mathbf{C}_t^{ij} = \int \phi_i(\mathbf{x}) \mathcal{S}_t \phi_j(\mathbf{x}) \mu(\mathbf{x}) \mathrm{d}\mathbf{x} \tag{9a}$$

$$\mathbf{r}_t^{j} = \int \phi_j(\mathbf{x}) \mathcal{S}_t \gamma(\mathbf{x}) \mu(\mathbf{x}) \mathrm{d}\mathbf{x}. \tag{9b}$$

We can then estimate the integrals in equation (9) from averages over an ensemble of simulations (that can be much shorter than the transition time, as mentioned above). Indexing simulations by $m = 1, 2, \ldots, M$,

$$\mathbf{C}_t^{ij} \approx \frac{1}{M} \sum_{m=1}^{M} \phi_i(\mathbf{X}^m(0)) \phi_j(\mathbf{X}^m(t \wedge T_{A \cup B})) \tag{10a}$$

$$\mathbf{r}_t^{j} \approx \frac{1}{M} \sum_{m=1}^{M} \phi_j(\mathbf{X}^m(0)) \gamma(\mathbf{X}^m(t \wedge T_{A \cup B})). \tag{10b}$$

Given these quantities, equation (8) can be solved for $\mathbf{v}$, which can be used to reconstruct $f$ through equation (7).

When the basis functions are indicator functions on sets of configurations, DGA is equivalent to using an MSM with appropriate boundary conditions to compute dynamical statistics. DGA can thus be viewed as a generalization of MSMs to arbitrary basis sets. As such, the data requirements for DGA can never be more than for MSMs, and they can be less if a basis better than indicator functions can be found[34,35]. As we discuss further in Basis set construction, we employ the MSM (indicator) basis set for computing the equilibrium probability, and a modified pairwise distance basis set[35] for the committor.

## State definitions

To perform committor calculations, we need to define which structures are in $A$ and $B$, or the down and up states, respectively. We defined the down state as

$$\left( \frac{d + 4.24\,\text{Å}}{1.1\,\text{Å}} \right)^2 + \left( \frac{\theta + 56.95°}{8°} \right)^2 < 1$$

and the up state as

$$\left( \frac{d + 0.506\,\text{Å}}{0.84\,\text{Å}} \right)^2 + \left( \frac{\theta - 3.94°}{7.6°} \right)^2 < 1,$$

where $d$ and $\theta$ are the S4 helix translocation and rotation, respectively. The centers and radii of the ellipses were obtained from several nanoseconds-long MD simulations initialized from the down and up states, rather than the crystal structures, since we wanted to include typical fluctuations. Visual inspection of the structures and trajectories showed that S4 helix movements often preceded or lagged behind the movements of arginine side chains, so we chose to include cutoffs on these distances as well. These are listed in Supplementary Table 1. To test for the possibility that our results depended on the choice of cutoffs, we altered the cutoffs and found that the qualitative features of the committor were preserved.

## Basis set construction

For the committor calculations, we used a basis set built from pairwise intramolecular distances[35]. The distances are based on two groups of residues: (1) R217, R223, R226, R229, and R232 and (2) D129, D136, D151,

D164, E183, and D186. We include all intergroup distances between the $C_\alpha$ atoms, as well as all distances between the arginine $C_\zeta$ atoms and the aspartate/glutamate $C_\gamma/C_\delta$ atoms of these residues. This yields $(5 \times 6) + (5 \times 6) = 60$ distances. We also tested including other pairwise distances, such as those involving residues in the hydrophobic plug, but these did not change the committor significantly.

Briefly, we constructed the basis functions by first computing the Euclidean distance for each configuration $\mathbf{x}$ to the boundary of state $A$ or state $B$, yielding $d_A$ and $d_B$ (the dependence on $\mathbf{x}$ is left implicit). Then, we define a function $h(\mathbf{x}) = d_A d_B / (d_A + d_B)^2$, which obeys the boundary conditions by construction, and set $\phi_i(\mathbf{x}) = \mathbf{x}_i h(\mathbf{x})$ where $\mathbf{x}_i$ is the $i$th component of the 60-dimensional coordinate vector. Finally, we use singular value decomposition to whiten the basis functions. The guess function is $\gamma(\mathbf{x}) = d_A^2 / (d_A^2 + d_B^2)$, which also satisfies the boundary conditions by construction.

For the weight factor, $w$, a basis of indicator functions (corresponding to an MSM) was used, since the distance basis tended to yield noisier results. Instead of the 60 salt-bridge distances, we used the minimum heavy-atom distance between each residue in the S4 helix and each residue in the S1−S3 helices (1924 distances in total). We reduced the feature vector to 10 dimensions using the integrated variational approach to conformational dynamics (IVAC)[49] and then clustered the MD structures with mini-batch $k$-means ($k = 200$), as implemented in scikit-learn 1.2.1[72]. Each cluster was converted into an indicator basis function.

## Simulation details and adaptive sampling procedure

The starting point for our simulations was the crystal structure of the up state (PDB ID 4G7V)[17]. We added hydrogens to this structure and embedded it into a palmitoyl oleyl phosphatidylcholine (POPC) lipid bilayer approximately 78 Å × 78 Å in area, with 86 and 88 lipids in the outer and inner leaflets, respectively. Titratable residues were assigned their default protonation states at pH 7. The orientation and relative position of the protein inside the membrane was adjusted according to the prediction from the orientations of proteins in membranes (OPM) database[73]. We then used VMD 1.9.4[74] to solvate the system in 0.1 M NaCl aqueous solution so that it was electrically neutral. There were 20 Na$^+$ and 19 Cl$^-$ ions, for a total of 56,582 atoms. The system setup is summarized in Supplementary Table 3.

For consistency with a previous simulation study of Ci-VSD[21], we used the CHARMM36m force field[75] for the protein, phospholipids, and ions, and the TIP3P model[76] for water molecules. We used the hydrogen mass repartitioning scheme[77] and a time step of 4 fs. All simulations were carried out in an isothermal-isobaric ensemble (300 K, 1 atm) with periodic boundary conditions and without an applied voltage. The temperature and pressure were controlled using the Nosé-Hoover thermostat and the semi-isotropic MTK barostat, respectively[78–80]. Long-range electrostatic interactions were calculated using the $u$-series method[81], and van der Waals interactions were truncated at a cutoff distance of 9 Å.

To construct a dataset that adequately sampled the transition, we employed a two-stage approach (Fig. 2). First, we drew initial conditions uniformly from equilibrated REUS windows in the space of S4 helix CVs[21]. The coordinates, corresponding velocities and periodic boundary parameters were used to initialize Anton 2 simulations[37]. From these initial conditions, we performed 230 simulations of 1 $\mu$s and 7 simulations of 10 $\mu$s for a total of 300 $\mu$s. Because these data were insufficient to converge the committor, we expanded our dataset by sampling adaptively. Specifically, we selected structures with estimated committor values near $q_+ = 0.5$ and with under-sampled values of the S4 helix and salt-bridge distance CVs.

In the initial dataset, we observed that some simulations did not diverge significantly from their initial structures on the timescale of hundreds of nanoseconds, suggesting that there were degrees of

freedom preventing fast equilibration of the desired CVs. Steering the system toward the the down or up state and back prior to running unbiased dynamics seemed to ameliorate this issue. These steered molecular dynamics simulations[82,83] were performed with a mild harmonic restraint on selected salt-bridge distances and on the root-mean-square deviation (RMSD) to either the down or up state. We used PLUMED 2.7[83] and AMBER20[84–86] to perform these simulations. From the final structures we initialized 115 additional 1 $\mu$s simulations on Anton 2. In total, we accumulated 415 $\mu$s of trajectories, with coordinates saved every 0.1 ns.

## Collective variables

We computed the intramolecular distances used as inputs to DGA using PyEMMA 2.5.12[87] and MDTraj 1.9.7[88]. Hydrogen bonds were computed with a donor-acceptor maximum heavy-atom distance of 3.5 Å and a minimum angle cutoff of 120° using the Hydro-genBondAnalysis module in MDAnalysis 2.4.2[89,90]. For salt-bridge distances between sensing arginines and acidic residues, we compute the distance between the $C_\zeta$ of arginine (the terminal carbon in the guanidinium group) and the $C_\gamma/C_\delta$ of aspartate/glutamate (the carbon in the carboxylate group). Between arginines and countercharges, we take the $N_\epsilon$ and both $N_\eta$ atoms of the guanidinium group as donors and $O_\delta/O_\varepsilon$ atoms of the carboxylate group in aspartate/glutamate as acceptors. For lipids, we choose the oxygens of the phosphatidylcholine headgroup in POPC as hydrogen bond acceptors.

The S4 helix translocation ($d$) and rotation ($\theta$) were computed with respect to the up state after aligning the $C_\alpha$ atoms of helices S1–S3 to their positions in the up state crystal structure so as to minimize the RMSD. The translocation was computed using the distanceZ function in the Colvars module of NAMD[91]. This function computes the distance along a projection of one vector onto another. In our case, the first vector was one connecting the centers of mass of $C_\alpha$ atoms of residues 217–233 in the structure of interest and the up state crystal structure, and the second vector was the principal inertial axis of these atoms in the up state crystal structure. The rotation was computed using the function spinAngle, measuring the angle of the rotation around the local helical axis of the $C_\alpha$ atoms of residues 217–233, with respect to the positions of these atoms in the up state crystal structure.

We computed the displacement charge, $Q_d$, according to the "Q-route"[38]:

$$Q_d = \sum_i q_i \left( \frac{z_i^{(u)} + L_z/2}{L_z} \right),$$ (11)

where $q_i$ are charges, $z_i^{(u)}$ are unwrapped $z$-coordinates (i.e., without applying periodic boundary conditions), and $L_z$ is the length of the simulation box along the $z$-axis (with the membrane parallel to the $xy$-plane).

## Sparse regression

To identify a small number of variables that are important for the dynamics, we modeled the committor using LASSO[44,92] (least absolute shrinkage and selection operator) as implemented in scikit-learn 1.2.1[72]. LASSO is a form of linear regression in which the loss is the sum of a least-squares term and a penalty term that encourages sparsity of the model coefficients $\beta_i$:

$$\mathcal{L}_{LASSO} = \sum_k \left( \beta^\top \mathbf{x}^{(k)} - q_+(\mathbf{x}^{(k)}) \right)^2 + \lambda \parallel \beta \parallel_1,$$ (12)

where $\|\beta\|_1 = \sum_i |\beta_i|$ is the $\ell_1$ norm and $\lambda$ is a hyperparameter controlling the penalty strength. In (12), $\mathbf{x}^{(k)}$ is a vector of features for the $k$th data point and $q_+(\mathbf{x}^{(k)})$ is the corresponding committor value. Prior to fitting, the data are standardized by removing the mean and normalizing to unit variance.

To overweight the transition region, where we expect the committor to be most informative, we sampled 100,000 points weighted by $\pi q_+(1-q_+)$, which has a maximum at $q_+ = 0.5$. As discussed in Results, our initial analysis suggested that the committor depends non-monotonically on some variables. To capture these relationships with LASSO, we split our dataset into points with $q_+ \leq 0.5$ and $q_+ \geq 0.5$ and then used an inverse sigmoid transform to map the DGA-estimated committor to the entire range $(-\infty, \infty)$ so that the LASSO-predicted committors were constrained to the appropriate range [either $(0, 0.5)$ or $(0.5, 1.0)$]. For the former, we chose the transformation $\ln[2q_+/(1 - 2q_+)]$, and, for the latter, we instead used the transformation $\ln[(2q_+ - 1)/(2 - 2q_+)]$. Although DGA can yield $q_+$ values outside the range $[0, 1]$, we clipped them to this range before applying the transformation.

As $\lambda$ varies, there is a tradeoff between sparsity and accuracy. Here, we chose $\lambda = 0.02$ for points $q_+ \leq 0.5$ and $\lambda = 0.03$ for points $q_+ \geq 0.5$ because they appeared to yield reasonably sparse representations while maintaining relatively good fits of the committor ($R^2 \approx 0.7$) given the linear form (Supplementary Fig. 11). Non-linear functions of the features, such as a polynomial or a neural network function, likely would better describe the committor. However, we used a linear model in the interest of maximizing interpretability.

## Reporting summary

Further information on research design is available in the Nature Portfolio Reporting Summary linked to this article.

## Data availability

The PDB files for the down and up states of Ci-VSD were obtained from wwPDB under accession codes 4G80 and 4G7V, respectively. The trajectory data without lipid or water coordinates, together with simulation parameters and initial and final structures of trajectories have been deposited in Zenodo under accession code 10.5281/zenodo.7502083. The raw simulation data have not been deposited due to their size; access can be obtained by contacting the authors or the Pittsburgh Supercomputing Center (PSC). Source data (CVs and results of DGA calculations) are provided with this paper.

## Code availability

Molecular dynamics simulation data were generated using the Anton 2 machine (code not publicly available; resource handled through the PSC) and AMBER20 with PLUMED 2.7. Analysis was performed using MDAnalysis 2.4.2, MDTraj 1.9.7, PyEMMA 2.5.12, scikit-learn 1.2.1, VMD 1.9.4, and IVAC along with custom scripts written in Python/Jupyter notebooks and tcl. A package for computing kinetic statistics with DGA is available at https://github.com/dinner-group/dgamem. Custom code is available at https://github.com/dinner-group/ci-vsd.

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

## Acknowledgements

The authors would like to thank Eduardo Perozo, Carlos Villalba-Galea, Adam Antoszewski, John Strahan, Chatipat Lorpaiboon, and Jonathan Weare for discussions and suggestions on the paper. S.C.G. acknowledges support from the National Science Foundation Graduate Research Fellowship under Grant No. 2140001. A.R.D. acknowledges National Institutes of Health (NIH) award R35 GM136381. B.R. acknowledges NIH/NIGMS award R01 GM062342. Anton 2 computer time was

provided by the Pittsburgh Supercomputing Center (PSC) through Grant R01 GM116961 from NIH. The Anton 2 machine at PSC was generously made available by D. E. Shaw Research. This work was completed in part with resources provided by the University of Chicago Research Computing Center and we are grateful for their assistance with the calculations. The "Beagle-3: A Shared GPU Cluster for Biomolecular Sciences" is supported by the NIH under the High-End Instrumentation (HEI) grant program award 1S10OD028655-0.

## Author contributions

S.C.G., B.R., and A.R.D. conceptualized the study. S.C.G. and R.S. performed the MD simulations. S.C.G. performed the dynamical analysis and regression. S.C.G., B.R., and A.R.D. interpreted the results and wrote the manuscript.

## Competing interests

The authors declare no competing interests.
