## [Peer Review File · Nature Communications]

Dynamics of activation in the voltage-sensing domain of *Ciona intestinalis* phosphatase Ci-VSPREVIEWER COMMENTS

Reviewer #1 (Remarks to the Author):

The manuscript by Guo et al. used molecular dynamics to elucidate the molecular mechanism of the resting-active transition for Ciona voltage-sensing phosphatase (Ci-VSP). The main findings in this paper are that the activation primarily involves the movement the arginine side chains and the exchange of salt bridges with negative charge cluster residues; and the movement of the arginine side chain is independent of the overall S4 movement involving rotation and translation.

General Comments:

This manuscript uses molecular dynamics to examine the extensively studied question of exactly how the S4 gating charges in voltage-gated ion channels move in response to membrane potential. There are two general concerns with this manuscript. First, the conclusions are at best incremental, providing details on how the gating charges move through the voltage sensor and interact with negative counter charges in the extracellular negative cluster of the voltage sensor. These charge-charge interactions are already well established by experiment. Second, the conclusions are not well supported by these simulations because of the short simulation times, in the range of hundreds of nanoseconds, that were used to predict gating charge movements on timescale of hundreds of microseconds, 1000-fold longer.

Specific Comments:

1. While I find the results obtained in this study interesting and seem to confirm the findings from many past studies about the activation mechanism of many voltage-sensing domains (VSDs) from different ion channels, I don't feel that the outcomes from this study have advanced our understanding of how voltage sensors operate in a significant way. The exchange of salt bridges between the arginine side chains and the side chains of glutamate or aspartate in the negative charge clusters has been proposed and verified by many structures of VSDs already. The movement of the charge on the guanidinium group of the arginine side chain is likely the main driving force for the S4 movement since the backbone has only partial charge from helical dipole and by itself should not be able to respond to the change in membrane potential in a meaningful way. Since burying an array of positive charges in the membrane is energetically unfavorable, the nature's solution is to have counter charges to interact, stabilize, and neutralize the positive charges of arginines in the membrane. So it makes sense that during the VSD activation especially when driven by membrane depolarization, the charge movement on the arginine side chains would occur and drive the translational and rotational movement of the entire S4 helix.

2. The experimental design in this study was to simulate the trajectories of the VSD without the phosphatase domain. While I find this approach reasonable as a systematic and well-focused study, it did not really capture what's going on in the cellular context. There are many unknowns regarding how VSD activate the phosphatase domain and the experimental design in this study unfortunately could not address this biologically important question.

3. Another concern about experimental design is that the simulations were carried out at 0 mV starting from either the Down– or the Up+ states and let the structure transition into their preferred states over the course of the simulations. As the authors pointed out, the simulations did not diverge significantly from the initial structures and steered MDs were needed to improve the issue. Also, Fig. 6b indicated that the energy landscape between the Down and Up states are very shallow. Would it be better and more biologically sound to carry out the simulations by holding the membrane at a hyperpolarized potential at the Ciona's resting membrane potential, then depolarize the membrane to 0mV for the activation, and do the opposite for the deactivation? The authors mentioned that mild hyperpolarizing or depolarizing potentials of ± 50 mV by adding a linear component QdV to the profile obtained under 0mV are sufficient to tilt the PMF in favor of the down and up states (Fig. 6b). It is unclear how this was done and it sounded like this is used only to calculate the PMF profile in Fig. 6b, but not for the main simulations.

4. Related to comment 3 above, I wonder how much the experimental design at 0 mV used in this study influences the finding about the translational movement of S4. The PMF from down to up states at 0 mV is very shallow with little energy barrier and could affect the movement of the arginine side chains/salt bridges vs. the translation of S4, so the lack of a single step or "click" movement found in other channels is prominent here.

5. I don't see how the authors concluded that the resting-active transition involves primarily the exchange of salt bridges independent of the S4 translocation. Fig 3d indicated to me that all the transitions happened between 250 and 500 ns with translocation and rotation plateaued at around ~500 ns around the same time when new salt bridges are formed by R226 and R229. The authors mentioned that vertical translocation frequently preceded or lagged behind the exchange of salt bridges. Fig. 6b seems to indicate that they all happen around the same time, which makes sense since the arginine side chains also need the backbone translocation to be able to reach the next negative charge residue.

6. This manuscript is very technical and will be difficult for the broad readership of Nature Communications to appreciate. The authors should revise the manuscript substantially to make it easier for the readers to understand or consider a different journal that is more suitable for the technical and theoretical nature of this paper.

Reviewer #2 (Remarks to the Author):

The study by Guo et al. is a technically excellent piece of work that uses new methods to analyze reaction pathways from ensembles of shorter simulations, and apply this to the gating transition of a voltage-sensor domain. There are many potential advantages of this method, and interesting analyses, but in terms of the application to voltage sensor gating it can be argued it primarily confirms the many previous studies (e.g. refs 7,16,17,18,19,20,25,29,30,31,32) rather than fundamentally changing our idea of the VSD gating. As the authors write, already in 2012 Jensen (ref. 32) showed an actual direct hyperpolarizing/depolarizing cycle of a complete voltage-gated ion channel, although admittedly with an elevated potential. One could of course argue whether inducing gating through an actual potential is more or less realistic than interpolating structures of the helix, but to this reviewer the key result is that the results are qualitatively similar, which per se is good.

1. While there are many things to be said for using ensemble sampling, the present approach is not entirely free of a priori assumptions about coordinates important for transition (page 3, bottom) since the simulations are started by drawing seeds from the space of collective variables used in a previous study based on umbrella sampling. This should be fine for getting seed structures of the helical backbone, but in particular during hyperpolarization the gating is driven by forces on the arginine side-chains. Since the gating process involves these side-chains crossing the hydrophobic plug and salt bridges exchanging, will the short simulations cover even a single such full event where salt bridges are broken, the residue moves across the plug, and salt bridges reformed? If not, don't we run a risk that we are missing some of the sampling/kinetics, and that this could explain e.g. why the authors perceive their transitions might be slower (which sounds reasonable if there is no potential acting on the arginine sidechain, in particular with the focused field around the plug)?

2. Given that that the present simulations start from seeds from the earlier umbrella sampling simulations in Ref 20, and the latter appears to reproduce the down-state free energy minimum better, what is the statistical reason why the present simulations are more accurate? As the authors write (bottom, page 5) a difference now observed *could* represent a previous unappreciated metastable state, but it could also be an artefact. So, is there any experimental or other independent support for it?

3. The claim on the bottom of page 6 about the observation of helix and sidechains being able to move independently as an argument against entire-s4-motion models seems like a bit of a strawman argument. I am not aware of a single study (in particular not any based on simulations) actually claiming that that S4+sidechains would move as a hard rigid body without any flexibility whatsoever. On the contrary, most studies rather seem to stress the importance of the potential primarily acting on the charged side chains, and there are numerous movies (including already in Jensen, 2012) showing how the sidechains need to be flexible when moving across the plug). Since there is no reference, is there any recent publication that has proposed such a fully-rigid-body model?

4. On page 8, the authors describe the non-monotonicity of committor vs. CVs, and how this forced them to separate the reaction in two parts, going from down and up states respectively, to the transition state. I can understand this from a technical point of view, but how are we then certain we sample the same clusters in the transition state, such that the whole transition is valid?

5. Possibly related to the previous point: We *know* based on the experimental up/down states that there is a large rotation of S4 that is central to the gating structure difference - and yet the rotation only has a relatively very low influence on the committor both from down-to-TS and TS-to-up. How is this compatible with the starting structures?

6. The free energy as a function of gating charge is a neat extension, and I fully agree with the authors that it is much more smooth, which is a key advantage. However, when looking at figure 4b, the actual free energy difference between down/up states is only $\sim 1\text{kT}$ without significant energy barriers, and even the down- or up+ states are only 2-2.5kT higher than down. This seems to predict that the voltage sensor should spend significant fractions of time on all four states even at 0mV, which seems to contradict experiments, where it is stable in either up or down states?

7. At the bottom of page 14, the authors derive an estimated timescale of the transition (2 microseconds) which appears to be much too fast (then it should be trivial to sample the entire gating in normal simulations). This is attributed to not taking barriers into the PMF into account, but looking at figure 4b again, such barriers aren't really present in the PMF, are they?

Reviewer #3 (Remarks to the Author):

In this paper the authors characterize the activation mechanism of the voltage-sensing phosphatase from *Ciona intestinalis* (Ci-VSP). The authors combine a large number of comparatively short unbiased trajectories and analyze them within the framework of the dynamical Galerkin approximation to compute potentials of mean force, committors and reactive currents. Moreover, a sparse linear fitting of the committor allows the identification of a small subset of variables important for the dynamics. The main finding of the work is that translation and rotation of helix S4 are not as tightly coupled as suggested by other simulation studies. In particular, the picture emerging from the paper is that activation occurs as a stepwise rearrangement of salt bridges more tightly coupled to rotation than to translation. Due to the sophisticated methodological approach and the far reaching implications for Voltage-gated ion channels, this referee recommends the publication of the paper provided the following points are addressed.

0) The manuscript describes an impressive simulation campaign which has been produced and analyzed with a set of innovative methods; this provides an indisputable methodological interest to the work, including the extensive dataset, which could prove useful for the community and should be made publicly available. However, the biophysical insights at which the work arrives do not appear in the current presentation as new as expected, e.g., as compared to previous work of some of the coauthors obtained with much simpler methods. The significance of the results may be easier to assess if their novelty was better highlighted, e.g., including a comparison with / proposition of experiments that could not be explained by previous approaches.

1) One of the greatest reasons of interest of the paper is the possibility to extend its findings to voltage gated ion channels. While in the Conclusion section the authors discuss differences and similarities of the activation mechanism of Ci-VSP with that of Kv1.2, their analysis of the literature is far from complete. Moreover, the citation of the work by Tao et al., "... the activation of Ci-VSD is qualitatively unlike

that of VGICs", might be hasty; new data have been possibly accrued since then. I therefore suggest to extend the review of the literature with a particular focus on computational works highlighting a loose dynamical coupling between S4 helix roto-translation and salt bridge exchange.

2) As the authors acknowledge, their methodological framework based on the dynamical Galerkin approximation (DGA) "... can be viewed as a generalization of Markov state models (MSMs)". Since the DGA method is highly original, while MSMs represent a well-established methodology, it would be useful if the authors expanded the discussion in the "Theory and methods" section to highlight similarity and differences between DGA and MSM. In particular, what are the advantages of the DGA method with respect to the MSMs? Also, could the authors comment with regards to the computational costs of the two approaches? The computational effort of the work presented by the authors (415 microseconds) is absolutely impressive. However, is this computational cost intrinsic to the DGA methodology? Would a more traditional approach based on MSMs be less expensive? This discussion would be of extreme relevance for anyone interested in using the same methodology.

3) The authors employ a sparse linear regression algorithm (LASSO) to fit the committor and identify a small subset of variables important for the activation dynamics. However, how could the author exclude some kind of non-linear dependence of the committor from the physical features chosen to describe the transition? Please comment on this.

4) The sentence at lines 211-215: "Although cation- π interactions and π - π stacking have been suggested to stabilize interactions between arginine and phenylalanine residues, such interactions are not explicitly included in the force field. Rather, R226 can make a salt bridge with D129 at the same time (Supplementary Fig 6c), and this interaction stabilizes the transition state." ought to be slightly

rephrased. Even if I understand what is meant, this sentence gives the wrong impression that no electrostatic interaction can be established between R226 and D129. The sentence should make clear that even if the partial charges on the guanidinium group of R226 and the phenyl ring of F161 allow for some electrostatic interaction, this does not accurately model cation- π and π - π interactions.

5) The sentence at lines 236-238

"By plotting the vector field of reactive flux on top of the PMF and committor (Fig 5c, arrows), we see that R226 moves towards D129 before R229 moves toward its position in the up state."

is not clear. In fact, as far as I understand, the position of R229 in the up state is in contact with D186, but Fig 5 only shows the R229-D129 distance. Am I missing something?

6) In Fig 6b the authors show the PMF as a function of the displacement charge. The PMF profile features several local minima corresponding to the down-, down, up, up+ states. Is there any relation between these minima and the peaks and valleys of the diffusion constant plotted in Supp. Fig. 9C? What are the steric/electrostatic hindrances limiting the motion of the displacement charge? Please comment on this.

7) At line 281 the authors use an approximate value of the diffusion constant (3×10^{-4}) e^2/ns to derive a diffusion timescale of the order of $2 \mu\text{s}$. However, the value $D = 3 \times 10^{-4} \text{e}^2/\text{ns}$ does not seem to match with the γ -scale of Supp. Fig. 9c where D ranges from 2.5×10^{-3} to 1.0×10^{-2} . Could you please double check? Also: how does the estimated timescale compare with experimental values?

8) In section "F. Basis set construction" the authors say that they compute the basis functions $\phi_i(x)$ using a vector of 60 distances ("This yields $(5 \times 6) + (5 \times 6) = 60$ distances"). This number however, does not match with the number of basic (R217, R223, R226, R229, R232) and acidic residues (D129, D136, D151, E183, D186) in the two groups chosen to compute pairwise intramolecular distances. Maybe there is a missing residue ? Please double check.

MINOR ISSUES

Please correct the misprints. The \AA Angstrom symbol should be capitalized in the following sentences.

a) Line 80 "... can translocate a few \aangstroms up".

b) line 196 "... can move up and down a few \aangstroms ..."

Point-by-point reply to reviewer comments

We thank the reviewers for their careful attention to our manuscript and helpful suggestions, which we have tried to address. Reviewer comments are reproduced below, and our replies follow in *italics*. Changes associated with Reviewer 1, 2, and 3's comments are marked in the manuscript with blue, red, and teal, respectively. Other changes are marked in violet.

REPLY TO REVIEWER 1

General comments

The manuscript by Guo et al. used molecular dynamics to elucidate the molecular mechanism of the resting-active transition for Ciona voltage-sensing phosphatase (Ci-VSP). The main findings in this paper are that the activation primarily involves the movement the arginine side chains and the exchange of salt bridges with negative charge cluster residues; and the movement of the arginine side chain is independent of the overall S4 movement involving rotation and translation.

This manuscript uses molecular dynamics to examine the extensively studied question of exactly how the S4 gating charges in voltage-gated ion channels move in response to membrane potential. There are two general concerns with this manuscript. First, the conclusions are at best incremental, providing details on how the gating charges move through the voltage sensor and interact with negative counter charges in the extracellular negative cluster of the voltage sensor. These charge-charge interactions are already well established by experiment. Second, the conclusions are not well supported by these simulations because of the short simulation times, in the range of hundreds of nanoseconds, that were used to predict gating charge movements on timescale of hundreds of microseconds, 1000-fold longer.

Response: *In contrast to previous studies of voltage-sensing proteins, which have focused mainly on metastable states of the voltage-sensing domain, our work dissects the microscopic events underlying a single "click." As we detail below, the mechanism that we describe represents a significant deviation from a canonical helical-screw model in that translocation, rotation, and side-chain movements are only loosely coupled. Our work also provides insight into the microscopic origin of differences in free energies of transition between Ci-VSP and voltage-gated ion channels (VGICs). These findings provide new insights into the origins of the complex kinetics of activation measured for wild type and mutant Ci-VSPs, which we have set in the context of results for other voltage-sensing proteins.*

Our conclusions arise from a theoretically sound framework based on transition path theory. The essential idea is that long-time statistics can be estimated by combining information from a dataset of short unbiased molecular dynamics trajectories that each samples a portion of the VSD activation event. This means that we can learn the mechanism of the transition even when no single trajectory connects the down and up states, so long as the transition region between them is well sampled. The approach both dramatically reduces the computational cost because we do not need to wait for fluctuations that give rise to such a trajectory, and increases the same statistical confidence. The kinetics that we obtain from our approach (presented in Section IIA, lines 200-217, in teal) are consistent with the reviewer's intuition that the transition takes hundreds of microseconds.

Specific comments

1. While I find the results obtained in this study interesting and seem to confirm the findings from many past studies about the activation mechanism of many voltage-sensing domains (VSDs) from different ion channels, I don't feel that the outcomes from this study have advanced our understanding of how voltage sensors operate in a significant way. The exchange of salt bridges between the arginine side chains and the side chains of glutamate or aspartate in the negative charge clusters has been proposed and verified by many structures of VSDs already. The movement of the charge on the guanidinium group of the arginine side chain is likely the main driving force for the S4 movement since the backbone has only partial charge from helical dipole and by itself should not be able to respond to the change in membrane potential in a meaningful way. Since burying an array of positive charges in the membrane is energetically unfavorable, the nature's solution is to have counter charges to interact, stabilize, and neutralize the positive charges of arginines in the membrane. So it makes sense that during the VSD activation especially when driven by membrane depolarization, the charge movement on the arginine side chains would occur and drive the translational and rotational movement of the entire S4 helix.

Response: *Although it is clear the positive charges on the arginine side chains must drive the motions of the S4 helix for the reasons stated, we emphasize that our study goes beyond explaining the energetic driving forces of activation. As noted above, our work dissects the microscopic events underlying a single "click", in contrast to previous studies of voltage-sensing proteins, which have focused mainly on metastable states of the voltage-sensing domain (refs. 25, 36, 39, 40), While there have been a number of computational studies of VSDs, the theoretical framework utilized here is critical to extract fine-grained dynamical information and its relation to microscopic features. Such analysis could not be addressed by previous studies.*

Despite capturing the apparent two-state behavior of the displacement charge, the calculations reveal multiple intermediates and deviations from a canonical helical-screw mechanism. In particular, activation is composed of several intermediate steps, involving sequential reorganization of the arginine side chains tied to a "slide latch"-like movement of the S4 helix backbone. Furthermore, we find that an acidic residue immediately above the hydrophobic plug (D129) can interact with arginines while they are in the plug, and this likely accounts for a markedly lower free energy barrier to transition than in VGICs. We have added the text at lines 98-106 and 419-424, and we have reorganized and rewritten the Results and Concluding Discussion to better emphasize the novelty of our findings.

2. The experimental design in this study was to simulate the trajectories of the VSD without the phosphatase domain. While I find this approach reasonable as a systematic and well-focused study, it did not really capture what's going on in the cellular context. There are many unknowns regarding how VSD activate the phosphatase domain and the experimental design in this study unfortunately could not address this biologically important question.

Response: *The objective of our study was to understand the mechanism of voltage sensing without the influence of other protein domains, which is challenging in VGICs, for example. This is a reductionist strategy to focus on the VSD itself. We agree with the reviewer that it would be interesting to study biological questions of how the voltage-sensing domain regulates the phosphatase domain in the future. The results that we obtain here provide the foundation for such studies.*

3. Another concern about experimental design is that the simulations were carried out at 0 mV starting from either the Down- or the Up+ states and let the structure transition into the their preferred states over the course of the simulations. As the authors pointed out, the simulations did not diverge significantly from the initial structures and steered MDs were needed to improve the issue. Also, Fig. 6b indicated that the energy landscape between the Down and Up states are very shallow. Would it be better and more biologically sound to carry out the simulations by holding the membrane at a hyperpolarized potential at the Ciona's resting membrane potential, then depolarize the membrane to 0 mV for the activation, and do the opposite for the deactivation? The authors mentioned that mild hyperpolarizing or depolarizing potentials of ± 50 mV by adding a linear component QdV to the profile obtained under 0 mV are sufficient to tilt the PMF in favor of the down and up states (Fig. 6b). It

is unclear how this was done and it sounded like this is used only to calculate the PMF profile in Fig. 6b, but not for the main simulations.

Response: *Although several simulations did not diverge significantly from the initial structures, we initialized our unbiased trajectories from each of the windows used for replica exchange umbrella sampling MD in ref. 25, which provided good coverage of configuration space between (and beyond) the down- and up+ states. A strength of our approach is that it can yield dynamical information at 0 mV without the need to drive conformational changes with (unphysiological) applied voltages as previous simulation studies did. However, the statistics that we obtain are for steady-state conditions. In principle, one could compute statistics for relaxation to from a perturbed initial condition, but this would involve additional assumptions.*

The energy landscape in Fig. 6b (now Fig. 3a) is computed with respect to the gating charge at 0 mV, and the alternative profiles at $\Delta V = \pm 50$ mV are obtained by adding the term $Q_d \Delta V$ term as stated. Here, Q_d is the displacement charge with respect to the down state, and the choice of the down state sets the zero of free energy. The displacement charge can be calculated as an average over the entire system in the absence of applied voltage. In the text we have clarified our procedure at lines 194-197.

4. Related to comment 3 above, I wonder how much the experimental design at 0 mV used in this study influences the finding about the translational movement of S4. The PMF from down to up states at 0 mV is very shallow with little energy barrier and could affect the movement of the arginine side chains/salt bridges vs. the translation of S4, so the lack of a single step or “click” movement found in other channels is prominent here.

Response: *We agree with the reviewer that the free-energy landscape obtained at 0 mV is relatively flat, and that applied potentials may alter the relative importance of different types of movements. This again points to the significant advance of our study relative to ones with applied potentials. Given that our results are consistent with available data and previous simulations on Ci-VSP, we believe that these results are accurate, and we now note at lines 451-455 that Ci-VSP may populate multiple states at a given voltage. As noted above, we discuss the microscopic origin of the low barrier relative to VGICs in the Concluding Discussion at lines 476-504 (in red and teal).*

5. I don’t see how the authors concluded that the resting-active transition involves primarily the exchange of salt bridges independent of the S4 translocation. Fig 3d indicated to me that all the transitions happened between 250 and 500 ns with translocation and rotation plateaued at around 500 ns around the same time when new salt bridges are formed by R226 and R229. The authors mentioned that vertical translocation frequently preceded or lagged behind the exchange of salt bridges. Fig. 6b seems to indicate that they all happen around the same time, which makes sense since the arginine side chains also need the backbone translocation to be able to reach the next negative charge residue.

Response: *In Fig. 3d (now Supplementary Fig. 6b), we show a single example of how the translocation of the S4 helix decouples from both the rotation and from the movements of the arginines. In particular, the time between 250 and 500 ns shows that the translocation (pink) increases steadily while the rotation lags behind (green), while the salt-bridge distances for R226 and R229 (bottom panel) remain almost fixed. However, we emphasize that this is just a single example and that the transition is stochastic. A strength of our analysis based on transition path theory is that it provides **statistical information** about the progress of the transition for the first time. Our conclusions are obtained by comparing the committor, the PMF, and the reactive current.*

6. This manuscript is very technical and will be difficult for the broad readership of Nature Communications to appreciate. The authors should revise the manuscript substantially to make it easier for the readers to understand or consider a different journal that is more suitable for the technical and theoretical nature of this paper.

Response: *We agree that the theoretical framework that we use, though well documented, is not very familiar to a large section of the community. We hope that the present application to the VSD can both showcase the method and serve as a pedagogical example. We have added an overview of the methods at the beginning of the Results section at lines 108-159 and in the Theory and Methods section at lines*

552-559 (in red) and 564-568. We now introduce the kinetic statistics and how they can be used to obtain mechanistic insights, and we explain heuristically how the conclusions reached about long-time behavior of the voltage sensor can be obtained from relatively short simulations. As noted above, we have reorganized the Results and Concluding Discussion to clarify our findings. Put together, we feel that the manuscript and the important insights that it provides about Ci-VSP should now be accessible to readers even if they are unfamiliar with simulation methods.

REPLY TO REVIEWER 2

The study by Guo et al. is a technically excellent piece of work that uses new methods to analyze reaction pathways from ensembles of shorter simulations, and apply this to the gating transition of a voltage-sensor domain. There are many potential advantages of this method, and interesting analyses, but in terms of the application to voltage sensor gating it can be argued it primarily confirms the many previous studies (e.g. refs 7,16,17,18,19,20,25,29,30,31,32) rather than fundamentally changing our idea of the VSD gating. As the authors write, already in 2012 Jensen (ref. 32) showed an actual direct hyperpolarizing/depolarizing cycle of a complete voltage-gated ion channel, although admittedly with an elevated potential. One could of course argue whether inducing gating through an actual potential is more or less realistic than interpolating structures of the helix, but to this reviewer the key result is that the results are qualitatively similar, which per se is good.

1. While there are many things to be said for using ensemble sampling, the present approach is not entirely free of a priori assumptions about coordinates important for transition (page 3, bottom) since the simulations are started by drawing seeds from the space of collective variables used in a previous study based on umbrella sampling. This should be fine for getting seed structures of the helical backbone, but in particular during hyperpolarization the gating is driven by forces on the arginine side-chains. Since the gating process involves these side-chains crossing the hydrophobic plug and salt bridges exchanging, will the short simulations cover even a single such full event where salt bridges are broken, the residue moves across the plug, and salt bridges reformed? If not, don't we run a risk that we are missing some of the sampling/kinetics, and that this could explain e.g. why the authors perceive their transitions might be slower (which sounds reasonable if there is no potential acting on the arginine sidechain, in particular with the focused field around the plug)?

Response: *As the reviewer notes, we chose the seed structures for the initial dataset based on the S4 helix coordinates rather than the arginine side-chains. However, during the adaptive sampling described in Theory and Methods we inspect other CVs (such as salt-bridge distances) which report on the arginine side-chain conformations. The advantage of our approach based on the committor is that it does not require the analysis to be focused around the CVs used for the initialization. We can project onto arbitrary combinations of CVs to investigate the relevance of physical coordinates to the transition, as presented in the manuscript.*

*Regarding the reviewer's question about sampling the full transition, we emphasize that the DGA framework used to compute kinetic and mechanistic statistics on the transition event does **not** require observation of a full transition path (i.e., a conformational change from the down state to the up state) or, say, even a full sequence of breaking one salt bridge and forming another. We only require that (overlapping) portions of the transition path are sampled (e.g., the breaking of a salt bridge and the formation of another could occur in different trajectories). Of course, DGA cannot learn the contribution of conformational changes that are not present in dataset, so care is required in constructing the dataset. Given that we see continuous coverage in all the projections that we analyze, we believe that we are not missing important intermediate conformations.*

To clarify these points, we have added text at the beginning of the Results at lines 116-134 (in blue) and in the Theory and Methods at lines 552-559 and 754-766.

2. Given that that the present simulations start from seeds from the earlier umbrella sampling simulations in Ref 20, and the latter appears to reproduce the down-state free energy minimum better, what is the statistical reason why the present simulations are more accurate? As the authors write (bottom, page 5) a difference now observed *could* represent a previous unappreciated metastable state, but it could also be an artefact. So, is there any experimental or other independent support for it?

Response: *We did not mean to imply that free energies from umbrella sampling are less accurate than those from DGA—we generally expect them to be more accurate. That said, an analogous minimum is also present in the REUS simulations at positive applied potentials in Fig. 1A of Ref. 25, when up-like states are stabilized. These observations lend support to the idea that a secondary metastable intermediate state exists between the down and up states. We have revised the discussion at lines 307-314 to clarify these points.*

3. The claim on the bottom of page 6 about the observation of helix and sidechains being able to move independently as an argument against entire-s4-motion models seems like a bit of a strawman argument. I am not aware of a single study (in particular not any based on simulations) actually claiming that that S4+sidechains would move as a hard rigid body without any flexibility whatsoever. On the contrary, most studies rather seem to stress the importance of the potential primarily acting on the charged side chains, and there are numerous movies (including already in Jensen, 2012) showing how the sidechains need to be flexible when moving across the plug). Since there is no reference, is there any recent publication that has proposed such a fully-rigid-body model?

Response: *It is true that previous studies (even structural studies) do not explicitly propose a fully rigid-body model for the movement of the S4 helix and its side chains. However, to our knowledge our study is the first to document and explain the significance of the intermediates between resting and active states. This mechanism represents a significant deviation from a canonical helical-screw mechanism in that translocation, rotation, and side chain movements are only loosely coupled. Existing simulation studies (most notably ref. 39, but also see refs. 36 and 40) do not resolve the events between “clicks” and appear to observe concerted transitions (e.g., Fig. 1C and 2B and Supplemental Movies 6 and 7 of ref. 39). The statistics of such events remain unclear because only a few events were observed even in the longest direct simulations, and they may reflect the large applied voltages.*

Our simulations show that the arginine side chains (R226, R229, and to some extent R232) alter their coordination quasi-independently and that this motion is not tightly coupled to S4-helix translocation and rotation. Interestingly, we found some support for this picture in the literature: simulations of Kv1.2 channel in refs. 33, 36 subject to hyperpolarizing voltage provide hints of a similar decoupling of rotation from side chain movements, and Fig. 1D in ref. 25 provides some indication of the ability for the S4 helix to translate while maintaining the arginine salt-bridges. We have edited our discussion at lines 328-345 and 469-486 to be more explicit about the differences between our findings and existing models of activation.

4. On page 8, the authors describe the non-monotonicity of committor vs. CVs, and how this forced them to separate the reaction in two parts, going from down and up states respectively, to the transition state. I can understand this from a technical point of view, but how are we then certain we sample the same clusters in the transition state, such that the whole transition is valid?

Response: *The separation of the transition into two stages is only for the LASSO regression to identify physically interpretable CVs. It is not related to the sampling of the dataset or to the calculation of the committors and reactive currents.*

5. Possibly related to the previous point: We *know* based on the experimental up/down states that there is a large rotation of S4 that is central to the gating structure difference - and yet the rotation only has a relatively very low influence on the committor both from down-to-TS and TS-to-up. How is this compatible with the starting structures?

Response: *Given that the rotation changes most rapidly close to $q_+ \approx 0$ and $q_+ \approx 1$, we do not think that separating the dataset for the LASSO limits the size of the rotation coefficient. Because the coefficient of one variable may be coupled to that of another, we mainly use the LASSO to identify important CVs and do not try to quantitatively interpret the coefficients. We have revised the discussion of the LASSO regression throughout the Results, both to reflect this perspective and to make the manuscript less technical in response to Reviewer 1’s comments.*

6. The free energy as a function of gating charge is a neat extension, and I fully agree with the authors that it is much more smooth, which is a key advantage. However, when looking at figure 4b, the actual free energy difference between down/up states is only $1 kT$ without significant energy barriers, and even the down- or up+ states are only 2-2.5 kT higher than down. This seems to predict that the voltage sensor should spend significant fractions of time on all four states even at 0mV, which seems to contradict experiments, where it is stable in either up or down states?

Response: *We now include the free energy along the estimated committor and displacement charge in Fig. 3, which shows a significant barrier between the down and up states. Despite its correlation with the committor, the gating charge is not as predictive of the dynamics, and thus the projection onto Q_d makes the free energy barriers appear lower. We now discuss the differences in the free energy profiles between the displacement charge and committor at lines 225-237 (in red) and 446-451.*

That said, we believe that it is likely that the voltage sensor (at least in the absence of other domains) occupies multiple states at 0 mV, and the relative populations shift upon hyperpolarization or depolarization. Electrophysiological measurements can only measure an ensemble average, which reflects the balance of the four states. The population of multiple states may contribute to the multi-exponential kinetics of fluorescence relaxation measured for labeled VSDs (refs. 17, 47-50), in addition to the intermediates that we identify through the free energy along the estimated committor. We have added a comment at lines 451-455 (in blue).

7. At the bottom of page 14, the authors derive an estimated timescale of the transition (2 microseconds) which appears to be much too fast (then it should be trivial to sample the entire gating in normal simulations). This is attributed to not taking barriers into the PMF into account, but looking at figure 4b again, such barriers aren’t really present in the PMF, are they?

Response: *There was in fact an error in our calculation of the diffusion constant, and we ultimately found that quantity difficult to interpret. We thus replaced it with more direct estimates of the relevant timescales that we can compute without reference to the PMF—the time-correlation of the displacement charge (Fig. 3b) and the mean-first passage time to the up state (Supplementary Fig. 2). These both suggest that activation occurs on the $\sim 100 \mu s$ timescale. We discuss the calculations of kinetics in more detail now at lines 201-218 (in teal).*

REPLY TO REVIEWER 3

In this paper the authors characterize the activation mechanism of the voltage-sensing phosphatase from *Ciona intestinalis* (Ci-VSP). The authors combine a large number of comparatively short unbiased trajectories and analyze them within the framework of the dynamical Galerkin approximation to compute potentials of mean force, committors and reactive currents. Moreover, a sparse linear fitting of the committor allows the identification of a small subset of variables important for the dynamics. The main finding of the work is that translation and rotation of helix S4 are not as tightly coupled as suggested by other simulation studies. In particular, the picture emerging from the paper is that activation occurs as a stepwise rearrangement of salt bridges more tightly coupled to rotation than to translation. Due to the sophisticated methodological approach and the far reaching implications for Voltage-gated ion channels, this referee recommends the publication of the paper provided the following points are addressed.

1. The manuscript describes an impressive simulation campaign which has been produced and analyzed with a set of innovative methods; this provides an indisputable methodological interest to the work, including the extensive dataset, which could prove useful for the community and should be made publicly available. However, the biophysical insights at which the work arrives do not appear in the current presentation as new as expected, e.g., as compared to previous work of some of the coauthors obtained with much simpler methods. The significance of the results may be easier to assess if their novelty was better highlighted, e.g., including a comparison with / proposition of experiments that could not be explained by previous approaches.

Response: *As discussed in response to Reviewer 1’s comments, our study dissects the microscopic events underlying a single “click,” in contrast to previous studies of voltage-sensing proteins, which have focused mainly on metastable states (e.g., refs. 25, 36, 39, 40). This has not been addressed by previous studies and may explain the differences in Ci-VSD compared with VGICs, such as the differing dependence of activation kinetics upon mutation of residues in the hydrophobic plug (see our response point 2). We now include proposed experiments (such as hindering the rotation of R226 by introducing an unnatural amino acid) to test the effects of specific residues implicated by our analysis. We have expanded the Concluding Discussion to discuss kinetic measurements (lines 446-455, in red and blue), how our results relate to mutation experiments (lines 487-504), and proposals for experiments (lines 508-519).*

2. One of the greatest reasons of interest of the paper is the possibility to extend its findings to voltage gated ion channels. While in the Conclusion section the authors discuss differences and similarities of the activation mechanism of Ci-VSP with that of Kv1.2, their analysis of the literature is far from complete. Moreover, the citation of the work by Tao et al., “... the activation of Ci-VSD is qualitatively unlike that of VGICs”, might be hasty; new data have been possibly accrued since then. I therefore suggest to extend the review of the literature with a particular focus on computational works highlighting a loose dynamical coupling between S4 helix roto-translation and salt bridge exchange.

Response: *As suggested by the reviewer, we have examined the literature further. As noted above in our response to Reviewer 2, point 3, we mention that some hints of loose coupling arise in simulations of a Kv1.2 channel (see refs. 33, 36). Ref. 25 also notes briefly that the presence of intermediate metastable states in the PMF correspond to shifts in the backbone of S4 (Figure 1D).*

Our analysis offers possible evidence for why activation of Ci-VSD could be qualitatively different than that of VGICs. In our simulations, we observe that R226 can make a salt bridge with D129 while in contact with the I126 and F161 side chains, and this facilitates R226’s movement through the hydrophobic plug. In VGICs, the site corresponding to D129 generally has a polar but uncharged amino acid (ref. 68), so an analogous interaction does not exist, leading to a significant dielectric barrier. The closer spacing of the acidic residues in Ci-VSD compared with VGICs may also explain how the arginine side chains can make salt bridges without the S4 helix needing to convert to a 3_{10} -helix near the hydrophobic plug, as in the sliding-helix mechanism (see also discussion in refs. 8, 25, 37). Perhaps relatedly, in VGICs, reducing the hydrophobicity of gating-pore residues by mutation accelerates activation upon depolarization (ref. 69), whereas in Ci-VSP changes in kinetics are more closely correlated with the sizes of side chains of hydrophobic-plug residues (refs. 28, 70). We have added these connections to the literature in the Concluding Discussion at lines 486-503.

3. As the authors acknowledge, their methodological framework based on the dynamical Galerkin approximation (DGA) “... can be viewed as a generalization of Markov state models (MSMs)”. Since the DGA method is highly original, while MSMs represent a well-established methodology, it would be useful if the authors expanded the discussion in the “Theory and methods” section to highlight similarity and differences between DGA and MSM. In particular, what are the advantages of the DGA method with respect to the MSMs? Also, could the authors comment with regards to the computational costs of the two approaches? The computational effort of the work presented by the authors (415 microseconds) is absolutely impressive. However, is this computational cost intrinsic to the DGA methodology? Would a more traditional approach based on MSMs be less expensive? This discussion would be of extreme relevance for anyone interested in using the same methodology.

Response: When the basis functions are indicator functions on sets of configurations, DGA is equivalent to using an MSM with appropriate boundary conditions to compute dynamical statistics. DGA can thus be viewed as a generalization of MSMs to arbitrary basis sets. As such, the data requirements for DGA can never be more than for MSMs, and they can be less if a basis better than indicator functions can be found. This comparison is made explicitly in Figs. 3-5 of ref. 41 and Fig. 5 of ref. 42. In the present work, we employ the MSM (indicator) basis set for computing the stationary distribution, and a modified pairwise distance basis set for the committor because we found those to perform best empirically. We have clarified these points at lines 660-670 in Theory and Methods.

4. The authors employ a sparse linear regression algorithm (LASSO) to fit the committor and identify a small subset of variables important for the activation dynamics. However, how could the author exclude some kind of non-linear dependence of the committor from the physical features chosen to describe the transition? Please comment on this.

Response: We do not exclude the possibility that non-linear functions of the features may better describe the committor, such as a polynomial or a neural network (which we tested in our work). However, we find that linear functions offer the greatest interpretability in terms of the coefficients—which is the ultimate goal of our committor analysis—without sacrificing too much accuracy. We have added a comment about our choice at lines 857-861.

5. The sentence at lines 211-215: “Although cation- π interactions and π - π stacking have been suggested to stabilize interactions between arginine and phenylalanine residues, such interactions are not explicitly included in the force field. Rather, R226 can make a salt bridge with D129 at the same time (Supplementary Fig 6c), and this interaction stabilizes the transition state.” ought to be slightly rephrased. Even if I understand what is meant, this sentence gives the wrong impression that no electrostatic interaction can be established between R226 and D129. The sentence should make clear that even if the partial charges on the guanidinium group of R226 and the phenyl ring of F161 allow for some electrostatic interaction, this does not accurately model cation- π and π - π interactions.

Response: We have edited the sentences at lines 372-379 to clarify that cation- π and π - π interactions are not treated accurately by the force field and that the stabilization of R226 in the plug is primarily due to an electrostatic interaction with D129.

6. “By plotting the vector field of reactive flux on top of the PMF and committor (Fig 5c, arrows), we see that R226 moves towards D129 before R229 moves toward its position in the up state.” is not clear. In fact, as far as I understand, the position of R229 in the up state is in contact with D186, but Fig 5 only shows the R229-D129 distance. Am I missing something?

Response: The reviewer is correct in noting that R229 is in contact with D186 in the up state, but we showed the projection onto the R229-D129 distance. In the revised version we have removed this plot and instead show the distributions (violin plots) of salt-bridge distances as a function of the committor in Fig. 6a. These correspond to distances which are physically-relevant during the down-up transition, and we feel that the representation is clearer and easier to interpret.

7. In Fig 6b the authors show the PMF as a function of the displacement charge. The PMF profile features several local minima corresponding to the down-, down, up, up+ states. Is there any relation between these minima and the peaks and valleys of the diffusion constant plotted in Supp. Fig. 9C? What are the steric/electrostatic hindrances limiting the motion of the displacement charge? Please comment on this.

Response: As noted above in response to Reviewer 2, point 7 and discussed further in response to the next comment, we now omit the calculation of the diffusion constant.

8. At line 281 the authors use an approximate value of the diffusion constant (3×10^{-4}) e²/ns to derive a diffusion timescale of the order of 2 μ s. However, the value $D = 3 \times 10^{-4}$ e²/ns does not seem to match with the y -scale of Supp. Fig. 9c where D ranges from 2.5×10^{-3} to 1.0×10^{-2} . Could you please double check? Also: how does the estimated timescale compare with experimental values?

Response: We thank the reviewer for pointing out this discrepancy: there was in fact an error in our calculation of the diffusion constant, making the timescale cited in the manuscript an order of magnitude too large. Because even with correction the diffusion constant was not straightforward to interpret, we have now included estimates of the timescales from the time-correlation of the displacement charge (Fig. 3b) and the mean-first passage time to the up state (Supplementary Fig. 2), both of which are on the order of 100 μ s. These values are about an order of magnitude faster than the fast component of fluorescence relaxation of dye-labeled VSDs, which tracks the charge movement (sensing current). However, the experimental studies were mainly performed with full-length Ci-VSP containing the phosphatase domain and linker, which slows down the voltage-activation process (ref. 47). These new calculations are discussed at lines 201-218.

9. In section “F. Basis set construction” the authors say that they compute the basis functions $\phi_i(x)$ using a vector of 60 distances (“This yields $(5 \times 6) + (5 \times 6) = 60$ distances”). This number however, does not match with the number of basic (R217, R223, R226, R229, R232) and acidic residues (D129, D136, D151, E183, D186) in the two groups chosen to compute pairwise intramolecular distances. Maybe there is a missing residue ? Please double check.

Response: The text was missing D164. We thank the reviewer for catching this mistake and have corrected it.

MINOR ISSUES

Please correct the misprints. The \AA Angstrom symbol should be capitalized in the following sentences.

- a) Line 80 “... can translocate a few \AA angstroms up”.
- b) line 196 “... can move up and down a few \AA angstroms ...”

By convention, when written out, the word “ \AA ngstrom” is not capitalized. See, e.g., <https://physics.nist.gov/cuu/Units/outside.html>.

REVIEWERS' COMMENTS

Reviewer #1 (Remarks to the Author):

This revised version of the manuscript has not addressed my primary concerns about this manuscript.

1. This work presents a modest advance at best. The motion of the gating charges and S4 segment in the voltage sensor domain are now well established by experiments in Ci-VSP and multiple ion channels. The details addressed here do not seem appropriate for Nature Communications.

2. This theoretical work analyzes transitions on the nanosecond time scale, but extrapolates from them to present a model for voltage sensor function on the millisecond time scale. It seems that this large extrapolation is not well justified.

3. The methods used here will not be easily appreciated by the broad readership of Nature Communications, and the incremental conclusions presented will likely not be of interest to most readers.

Reviewer #3 (Remarks to the Author):

The authors have satisfactorily addressed all the issues raised by this Reviewer and they exploited my remarks to significantly improve the paper that now better explains the significance of their work and is much more readable. The only minor flaw that still persists is a systematic error in the citation of Supplementary Figures. A few examples are the following:

line 185:

'... discrete values (Supplementary Fig ??a), ...'

lines 207-208:

'... down and up states ($\sim 100 \mu\text{s}$, Supplementary Fig ??).'

lines 489-490:

'... the I126 and F161 side chains (Supplementary Fig ??c).'

line 526:

'... transition state (Supplementary Fig ?? b).'

After fixing this typesetting issue, I have no reservation in warmly recommending the publication of this work.

Reviewer #4 (Remarks to the Author):

This paper studied microscopic details of translation and rotation of S4 helix of one-click motion based on the previously solved two X-ray crystal structures of Ci-VSD through a novel method of MD simulation. Authors did committer-analysis of multiple unbiased calculations during the course of entire structural change from resting state (down) to activated state (up). Data were analyzed using DGA method, which is a dilated version of Markov model, and determined structural change of translation and rotation of S4 as the function of Qd, then PMF during resting to activated state was determined by this Qd. Further they analyzed contribution of salt bridges between acidic residues and basic residues during structural change from resting to activated state through LASSO regression method and CVs were determined.

Conclusions have confirmed previously proposed models of motion of S4 and there is not much novelty in biological findings. On the other hand, this work showed first example of detailed microscopic events of S4 providing insights into kinetics of voltage sensor motion through using novel method. This is enthusiastic by itself. I believe that this work also suggests one of future directions of research on dynamics of membrane proteins more than voltage sensor. Unfortunately, this paper is difficult to read in particular by people who are beginners of MD simulation and therefore not suitable for Nature communications at least in this version. I therefore strongly recommend to revise the paper by extensive rewriting. Detailed comments are below.

1. Results showed that voltage sensor takes several microscopic states even at 0 mV in Ci-VSD. This is very interesting. On the other hand, readers would like to know how results are altered if membrane voltage is applied (depolarization or hyperpolarization). Authors mentioned in discussion that this is future plan, but it will be better to mention the reason why analysis with negative or positive membrane potential could not be performed in this paper (it may perhaps require too much load of calculation). Authors can explain how this paper is novel and of significance even though calculation was performed at 0 mV steadystate.
2. Abstract would be rewritten so that novel method to study dynamics of membrane proteins is the main point of this paper. Then, abstract would say that such novel methodology has been applied to a

simple model system of voltage sensing function, or Ci-VSD, etc. Accordingly, probably the title of the paper would be changed so that readers could guess that this study is done in silico not by experiments. Current version of title seems to make readers to imagine that this paper is addressing dynamics of VSD motion by electrophysiological measurement, not by MD simulation.

3. Showing summary figure in Fig7 is good. I thank authors for including this and I guess that authors did their best to make this figure. But I am afraid that it is still difficult to understand by readers of Nature communications. For example, I do not fully understand where numbering of amino acid residue and orange circle correspond to in the helices. In this cartoon, as speculated from the main text, relative position of hydrophobic plug and salt bridge is probably one of the key conclusions. But this is not easily seen in Fig.7. Please consider any way of improving transferability of results to the important concepts for example by making another figure or another video.

4. It may be helpful to pay attention to the paper by Mizutani N. et al. (PNAS, 119 (26), e2200364119, 2022) which studied the extent of motion of S4 by quenching of fluorescence introduced to S4 by tryptophan 182, the landmark on S3.

5. In “Concluding Discussion”, it is said that R226’s movement through the hydrophobic plug is facilitated by its ability to make a salt bridge with D129. Any previous reference with mutagenesis of D129 in VSD motion would be cited and discussed in more detail. One such reference is probably PMC3699739, but other reference with more detailed analysis might be available.

Reviewer #5 (Remarks to the Author):

This study by Guo et al. is extremely innovative from the methodological point of view and provides valuable insight into the fundamental biophysical problem of voltage sensitivity. Although many of the findings from this work are consistent with molecular mechanisms already described in literature, the kinetic modeling is entirely new and provides valuable insight and experimentally testable predictions. I find that the authors made a good job at revising the paper and I do not have additional revisions to suggest. All technical aspects of the calculations are sound and the conclusions drawn from them are appropriate. The manuscript is well written and accessible to a wide audience. Overall, I think that these results are sufficiently novel and fundamental to be of interest to a large community of scientists. I strongly recommend publication of the current version of the manuscript.

REPLY TO REVIEWER #1

This revised version of the manuscript has not addressed my primary concerns about this manuscript.

1. This work presents a modest advance at best. The motion of the gating charges and S4 segment in the voltage sensor domain are now well established by experiments in Ci-VSP and multiple ion channels. The details addressed here do not seem appropriate for Nature Communications.
2. This theoretical work analyzes transitions on the nanosecond time scale, but extrapolates from them to present a model for voltage sensor function on the millisecond time scale. It seems that this large extrapolation is not well justified.
3. The methods used here will not be easily appreciated by the broad readership of Nature Communications, and the incremental conclusions presented will likely not be of interest to most readers.

Response: Our computational framework has been carefully vetted and benchmarked in previous peer reviewed publications, so we are confident that it is theoretically sound. We agree with the reviewer that it is truly remarkable that the framework yields long-time statistics from short-trajectory data. While we note that the other reviewers were able to appreciate the novelty of our work, we have edited the abstract to clarify the previous state of knowledge and our contributions, and we have carefully reviewed the remainder of the manuscript to remove unnecessary jargon.

REPLY TO REVIEWER #3

The authors have satisfactorily addressed all the issues raised by this Reviewer and they exploited my remarks to significantly improve the paper that now better explains the significance of their work and is much more readable. The only minor flaw that still persists is a systematic error in the citation of Supplementary Figures. A few examples are the following: line 185: '... discrete values (Supplementary Fig ??a), ...'

lines 207-208: '... down and up states ($100 \mu\text{s}$, Supplementary Fig ??).'

lines 489-490: '... the I126 and F161 side chains (Supplementary Fig ??c).'

line 526: '... transition state (Supplementary Fig ?? b).'

After fixing this typesetting issue, I have no reservation in warmly recommending the publication of this work.

Response: We are glad the reviewer found the paper significantly more readable and felt that we had resolved all the issues previously raised. We thank them for noticing the issues with the references to the supplementary figures and have fixed them.

REPLY TO REVIEWER #4

This paper studied microscopic details of translation and rotation of S4 helix of one-click motion based on the previously solved two X-ray crystal structures of Ci-VSD through a novel method of MD simulation. Authors

did committer-analysis of multiple unbiased calculations during the course of entire structural change from resting state (down) to activated state (up). Data were analyzed using DGA method, which is a dilated version of Markov model, and determined structural change of translation and rotation of S4 as the function of Qd, then PMF during resting to activated state was determined by this Qd. Further they analyzed contribution of salt bridges between acidic residues and basic residues during structural change from resting to activated state through LASSO regression method and CVs were determined.

Conclusions have confirmed previously proposed models of motion of S4 and there is not much novelty in biological findings. On the other hand, this work showed first example of detailed microscopic events of S4 providing insights into kinetics of voltage sensor motion through using novel method. This is enthusiastic by itself. I believe that this work also suggests one of future directions of research on dynamics of membrane proteins more than voltage sensor. Unfortunately, this paper is difficult to read in particular by people who are beginners of MD simulation and therefore not suitable for Nature communications at least in this version. I therefore strongly recommend to revise the paper by extensive rewriting. Detailed comments are below.

1. Results showed that voltage sensor takes several microscopic states even at 0 mV in Ci-VSD. This is very interesting. On the other hand, readers would like to know how results are altered if membrane voltage is applied (depolarization or hyperpolarization). Authors mentioned in discussion that this is future plan, but it will be better to mention the reason why analysis with negative or positive membrane potential could not be performed in this paper (it may perhaps require too much load of calculation). Authors can explain how this paper is novel and of significance even though calculation was performed at 0 mV steadystate.

Response: We expect our results to hold at moderate applied potentials and already examine the effects of mildly depolarizing and hyperpolarizing voltages on the potential of mean force as a function of the displacement charge (Fig. 3). To compute thermodynamics and kinetics beyond the linear response regime would require sampling under applied potentials, which is currently prohibitively computationally costly.

2. Abstract would be rewritten so that novel method to study dynamics of membrane proteins is the main point of this paper. Then, abstract would say that such novel methodology has been applied to a simple model system of voltage sensing function, or Ci-VSD, etc. Accordingly, probably the title of the paper would be changed so that readers could guess that this study is done in silico not by experiments. Current version of title seems to make readers to imagine that this paper is addressing dynamics of VSD motion by electrophysiological measurement, not by MD simulation.

Response: We have edited the abstract to clarify that our contribution is the application of our recently developed computational framework.

3. Showing summary figure in Fig7 is good. I thank authors for including this and I guess that authors did their best to make this figure. But I am afraid that it is still difficult to understand by readers of Nature communications. For example, I do not fully understand where numbering of amino acid residue and orange circle correspond to in the helices. In this cartoon, as speculated from the main text, relative position of hydrophobic plug and salt bridge is probably one of the key conclusions. But this is not easily seen in Fig.7. Please consider any way of improving transferability of results to the important concepts for example by making another figure or another video.

Response: We have added additional text to the caption to specify that the sticks and the colored circles correspond to specific residues.

4. It may be helpful to pay attention to the paper by Mizutani N. et al. (PNAS, 119 (26), e2200364119, 2022) which studied the extent of motion of S4 by quenching of fluorescence introduced to S4 by tryptophan 182, the landmark on S3.

Response: We thank the reviewer for this suggestion. W182 may be a potential target for mutagenesis in understanding the kinetics of Ci-VSD. We now note this with the suggested reference at line 346.

5. In “Concluding Discussion”, it is said that R226’s movement through the hydrophobic plug is facilitated by its ability to make a salt bridge with D129. Any previous reference with mutagenesis of D129 in VSD motion would be cited and discussed in more detail. One such reference is probably PMC3699739, but other reference with more detailed analysis might be available.

Response: We have added the suggested reference and an additional reference to the voltage-gated proton channel Hv1, which also possesses a conserved aspartate in a corresponding sequence position. Mutagenesis experiments do indeed indicate that this residue changes the voltage-dependence of VSD motion, as suggested. We now include a discussion of these experiments at lines 332-333.

REPLY TO REVIEWER #5

This study by Guo at al. is extremely innovative from the methodological point of view and provides valuable insight into the fundamental biophysical problem of voltage sensitivity. Although many of the findings from this work are consistent with molecular mechanisms already described in literature, the kinetic modeling is entirely new and provides valuable insight and experimentally testable predictions. I find that the authors made a good job at revising the paper and I do not have additional revisions to suggest. All technical aspects of the calculations are sound and the conclusions drawn from them are appropriate. The manuscript is well written and accessible to a wide audience. Overall, I think that these results are sufficiently novel and fundamental to be of interest to a large community of scientists. I strongly recommend publication of the current version of the manuscript.

Response: We thank the reviewer for their positive comments.